



# Benchmark campaign and case study episode in Central Europe for development and assessment of advanced GNSS tropospheric models and products

J. Douša[1], G. Dick[2], M. Kačmařík[3], R. Brožková[4], F. Zus[2], H. Brenot[5], A. Stoycheva[6], G. Möller[7], J. Kaplon[8]

[1]Geodetic Observatory Pecný, Reasearch Institute of Geodesy, Topography and Cartography, Zdiby, 250 66, Czech Republic
[2]Helmholtz Centre Potsdam - GFZ German Research Centre for Geosciences, Potsdam, D-14473, Germany
[3]Institute of Geoinformatics, VŠB – Technical University of Ostrava, Ostrava, 70 833, Czech Republic
[4]Numerical Wheather Prediction Department, Czech Hydrometerological Institute, Praha, 143 06, Czech Republic
[5]Atmospheric Composition Department, Royal Belgian Institute for Space Aeronomy, Brussels, B-1180, Belgium
[6]Forecasts Department, National Institute of Meteorology and Hydrology, Sofia, 1784, Bulgaria
[7]Department of Geodesy and Geoinformation, TU Wien, Vienna, 1040, Austria
[8]Institute of Geodesy and Geoinformatics, Wroclaw University of Environmental and Life Sciences, Wroclaw, 50-357, Poland

*Correspondence to*: J. Douša (jan.dousa@pecny.cz)

**Abstract.** Initial objectives and design of the Benchmark campaign organized within the European COST Action ES1206 (2013-2017) are described in the paper. This campaign has aimed at supporting the development and validation of advanced GNSS tropospheric products, in particular high-resolution and ultra-fast zenith total delays (ZTD) and tropospheric gradients derived from a dense permanent network. A complex dataset was collected for the 8-week period when several extreme heavy precipitation episodes occurred in central Europe which caused severe river floods in this area. An initial processing of data sets from Global Navigation Satellite System (GNSS) and numerical weather models (NWM) provided independently estimated reference parameters – zenith tropospheric delays and tropospheric horizontal gradients. Their provision gave an overview about the product similarities and complementarities and thus a potential for improving a synergy in their optimal exploitations in future. Reference GNSS and NWM results were inter-compared and visually analysed using animated maps. ZTDs from two reference GNSS solutions compared to global ERA-Interim re-analysis resulted in the accuracy at the 10-millimeter level in terms of RMS (with a negligible overall bias), comparisons to global GFS forecast showed accuracy at the 12-millimeter level with the overall bias of -5 mm and, finally, comparisons to meso-scale ALADIN-CZ forecast resulted in the accuracy at the 8-milllimetre level with a negligible total bias. The comparison of horizontal tropospheric gradients from GNSS and NWM data demonstrated a very good agreement among independent solutions with negligible biases and the accuracy of about 0.5 mm. Visual comparisons of maps of zenith wet delays and tropospheric horizontal gradients showed very promising results for future exploitations of advanced GNSS tropospheric products in meteorological applications such as severe weather event monitoring and weather nowcasting. The GNSS products revealed a capability of providing more detailed structures in atmosphere than the state-of-the-art numerical



weather models are able to capture. Initial study on contribution of hydrometeors (e.g. cloud water, ice or snow) to GNSS signal delays during severe weather reached up to 17 mm in zenith path delay and suggested to carefully account them within the functional model. The reference products will be further exploited in various specific studies using the Benchmark dataset. It is thus going to play a key role in these highly inter-disciplinary developments towards better mutual benefits from

advanced GNSS and meteorological products.

## 1 Introduction

Microwave signal from Global Navigation Satellite Systems (GNSS) necessarily intersects earth atmosphere when propagating from medium orbit satellites to a ground-based receiver. According to the GNSS signal frequencies of 1-2 GHz, the lower part of the atmosphere, i.e. troposphere, is a non-dispersive medium which delays all the signals in the same way.

This means there is no way of eliminating the effect due to the troposphere in analyses using GNSS observations only. The tropospheric effect can be, however, precisely calculated by determining the refractive index of the atmosphere along the signal path which is a function of the atmospheric pressure, temperature, humidity and empirical constants, see Appendices A and B.

In zenith direction above the receiver, the signal mean delay due to the troposphere reaches 2.3 meters at mean sea level and

decreases generally with increasing altitude of the receiver. The signal path delay, however, increases with the secant of the zenith distance of satellite and reaches up to tens of meters close to the horizon. A synergy between GNSS and meteorological observations and products became important together with increasing demands on the accuracy of GNSS positioning applications on the one hand, and a fast development in numerical weather forecasting on the other hand.

The concept of the ground-based GNSS-meteorology was introduced by Bevis et al. (1992). Various projects aimed at

developing and evaluating the highly effective and complementary method for sounding of water vapour in the atmosphere in support of weather forecasting have been completed. In 2000-2001, the concept was successfully implemented and tested by several groups in Europe (Gendt et al., 2001; Douša, 2001) for an operational production initiated during the COST Action 716 Demonstration campaign (Elgered et al., 2005). Since 2005, the operational production of tropospheric delays from the ground-based GNSS stations is coordinated and monitored by the EUMETNET EIG GNSS Water Vapour

Programme (E-GVAP, 2005-2017, Phase I-III, http://egvap.dmi.dk).

Developments of GNSS-meteorology in Europe have helped to establish a close collaboration between the meteorological and geodetic communities over two decades. The current operational GNSS data processing in support of meteorological applications uses mainly the US NAVSTAR Global Positioning System (GPS). The tropospheric path delays are usually estimated in zenith direction providing an hourly update rate and the same temporal resolution. Fast developments in both

GNSS and meteorology domains during the past decade provided an excellent opportunity for enhancing the synergy via developing new products and applications on both sides.



On the GNSS side, we should specifically mention the following: a) availability of global data and a fast progress in processing data from all global satellite multi-constellation (GPS, GLONASS, Galileo, BeiDou), b) availability of real-time (RT) data and analysis tools, c) availability of global orbit and clock real-time products (Caissy et al., 2012) in support of autonomous Precise Point Positioning, PPP (Zumberge et al., 1997), d) improved tropospheric (and other) models and products in long-term homogeneous data re-processing. Besides improving the accuracy of GNSS-derived tropospheric parameters for use in meteorology, the abovementioned developments enables also enhancements in terms of aspects such as: a) monitoring of an anisotropy of the troposphere around GNSS stations, b) providing a higher temporal resolution and an ultra-fast update of parameters exploiting highly efficient and autonomous PPP method, and c) reconstructing vertical water vapour distribution using tomography approach.

On the meteorological side, a progressive increase in spatial and temporal resolutions of numerical weather models (NWM), the forecast update rates and developments in numerical and non-numerical weather nowcasting and severe weather event monitoring requires more high-quality observations (Bauer et al., 2015), in particular water vapour as a key atmosphere constituent in the weather modelling. On the other hand, data from steadily enhanced NWMs, both operational forecasting and re-analyses, are of interest in other applications including positioning and navigation using space geodetic techniques such as GNSS.

In order to stimulate and coordinate developments and assessments of next-generation GNSS tropospheric products and applications aimed at improving the quality of precise positioning, weather forecasting or climate monitoring, the new EU COST Action ES1206 "Advanced Global Navigation Satellite Systems tropospheric products for monitoring severe weather events and climate (GNSS4SWEC)" was launched for the period of 2013–2017. Three working groups (WG) were established in order to cover main domains of interest specified in WG names:

- WG1: Advanced GNSS processing techniques
- WG2: GNSS for severe weather monitoring
- WG3: GNSS for climate monitoring

Close co-operations among these working groups were envisaged from the beginning in order to reach an optimal synergy of new data, products and developed applications. The COST Action aims at supporting an effective coordination of scientific developments in Europe and thus ten topics were identified for collaborations within WG1 domain. A need then arose for an effective sharing of data, products, tools, and know-how in order to stimulate and fasten the progress. The idea of preparing a common Benchmark campaign in support of WG1 developments, considering also a close co-operation with activities in WG2 and WG3, was motivated by two principles:

- To collect a complete, unrivalled, pre-processed and cleaned, properly documented data set in support of development and assessment of enhanced tropospheric models and products for both GNSS-meteorology and GNSS precise positioning.



- To enable an effective collaboration in small groups for complementary topics within the COST ES1206 Action.

The aim of this paper is to introduce the GNSS4SWEC Benchmark campaign, the data set and reference products, designed for a collaborative development towards improving tropospheric models and products in GNSS meteorology and precise positioning. Section 2 introduces and describes the Benchmark campaign design. Section 3 characterizes flood episode in central Europe in June 2013 which was selected for a particular interest in study of extreme weather event. Section 4 provides a description of complex data set collected for the Benchmark. Section 5 summarizes results of initial analyses and reference products in support of further collaboration using the campaign. The summary and scheduled activities using the data from the Benchmark campaign are given in Section 6. The paper is completed with Appendices A and B describing the GNSS tropospheric model and its functional relation to meteorological parameters.

## 2 GNSS4SWEC Benchmark campaign

The planning of the Benchmark campaign consisted of three steps. Firstly, an inventory of requirements, interests and data sets was performed. Secondly, data were collected, documented and initially processed including the data quality checking and data cleaning. Thirdly, specific goals and activities were defined and their coordination was planned. Results from specific activities using the Benchmark data set are expected until June 2017, i.e. the end of the COST Action.

### 2.1 Description of WG1 objectives

The motivation for the Benchmark design is related to the main goals of the WG1 in GNSS4SWEC project, which are the following:

- Coordinating development of advanced tropospheric products in support of weather forecasting, namely ultra-fast products, asymmetry monitoring, tomography enhancement and multi-constellation processing; all optimally supported with the autonomous PPP processing method.

- GNSS data reprocessing and assessment of involved models in order to provide consistent tropospheric products for climate research.

- Exploiting numerical weather data in precise GNSS retrievals and validation which includes a) assessing and improving mapping of tropospheric delays from slant directions to the zenith, b) improving a priori models for tropospheric effect and the separation of hydrostatic and non-hydrostatic parts of the effect, c) retrieving and evaluating tropospheric horizontal gradients, d) developing and evaluating tropospheric correction models for real-time applications (positioning, navigation and other GNSS parameter retrievals).

- Stimulating transfer of knowledge, tools and data exchange in support of new analysis centres and networks setup.



In order to efficiently lead and coordinate all the foreseen work within specified goals ten sub-working groups were defined, each of them consisting of contributions from several research institutions. Besides, a specific sub-working group was set to support an organization of the Benchmark campaign.

## 2.2 Data inventory and requirements for the Benchmark design

In prior of planning the Benchmark campaign, an inventory of requirements from individual WG1 topics was provided in terms of a) the campaign period and season, b) area of interest and domain size, c) weather conditions, d) request and limitations for number of stations and their density in the area, e) requests on data types and their availability, f) inventory of tools, software and facilities available for the campaign, and g) interest of WG1 members on contributions and coordination tasks.

Although the requirements of various groups were not always in a full agreement, the following requests for the Benchmark data set were summarized:

- Period covering a month at least to enable NWM and GNSS processing initializations and to cover different weather conditions – quiet and variable, optimally including a severe weather event.

- Availability of a dense network of GNSS reference stations with a limited scale, but including flat and mountainous areas, optimally not far from the centre of Europe.

- Availability of meteorological data, in particular atmospheric pressure, partial water vapour pressure and temperature. These should come, optimally, from independent sources such as synoptic sites, radiosonde profiles, numerical weather models or other water vapour observing techniques, e.g. Water Vapour Radiometer (WVR).

- In order to fulfil all requirements, the data set is considered to be completed as a whole while various sub-sets will be defined and used for specific activities.

## 2.3 Selected spatial and temporal domain

The area in central Europe covering Germany, the Czech Republic, Poland and Austria was selected for the Benchmark campaign suiting a majority of requirements as it concerned WG1 topics. Initially, the territory in North-West Europe around Belgium and the Netherlands was considered as more interesting for the study of severe weather events in WG2 mainly due to stronger and more frequent weather front systems. However, it is a flat area and, due to other limitations, the Benchmark campaign was preferably designed to serve mainly WG1 activities such as developing, optimizing and assessing new strategies, all under well-known and documented conditions and data sets. The WG1 optimized solutions will be then ready for use in severe weather event study cases collected by WG2 for different areas and periods.



The period of two months was selected for the campaign – May and June, 2013. The weather conditions in the selected area during May 2013 were mostly quiet, however at the beginning and during June a few events of heavy raining occurred from which some were partly captured by most of the numerical weather models, details in Sect. 0.

The domain selected for the campaign finally covered a larger region that was split into two parts, see Figure 1. Additionally,
seven clusters were set for an effective GNSS data processing. The first and the most important part, labelled as a 'core' domain, covered areas hit by extreme precipitation events and floods during June 2013, see yellow area in Figure 1. It extends from South-East Germany across South-West Poland and the Czech Republic to North-West Austria. The second part of the campaign, labelled as an 'extended' domain, covered remaining parts of Germany and North-West Poland.

## 2.4 Envisioned studies and activities

Based on the goals and motivations of WG1 and its sub-working groups, a list of topics supported by the Benchmark dataset has been prepared. Since an exploitation of the data set is foreseen to last up to two years, the list will be continuously updated. A brief summary of currently envisioned tasks and goals for the campaign is thus provided:

- Assessment of real-time (RT) and near real-time (NRT) tropospheric products in support of meteorological applications – development of optimal strategies, evaluating new analytical centres, study of the impact of multi-
GNSS constellations.

- Improved GNSS tropospheric modelling for positioning using NWM data fields – exploitation of NWM data for improved GNSS positioning both in post-processing and in real-time positioning and navigation.

- Development and assessment of advanced products of asymmetry modelling and monitoring of the troposphere – developing and optimizing strategies for a production of tropospheric horizontal gradients and slant delays, and
comparison to WVR and NWM data.

- Estimation and exploitation of GNSS tropospheric horizontal gradients for meteorological applications – study of the potential of utilization of GNSS horizontal gradients in numerical or non-numerical weather forecasting and nowcasting, understanding of higher-order gradients and contributions from hydrostatic and wet parts.

- Support for RT and NRT separation of ZHD/ZWD and IWV map production – development of ZTD to IWV
conversions for RT and NRT scenarios, assessment of uncertainty of various meteorological data input (NWM, synoptic stations, and in-situ observations), and development of optimal interpolation strategies.

- Implementation of synthetic tropospheric products (zenith total delays, horizontal gradients, slants delays and residuals) and comparison with observations – assess the quality of NWMs for tropospheric correction models and its dependence on the prediction length. Preparation of reference data for testing in weather nowcasting applications
and tomography.



## 3 Case study episodes in 2013

In this section we offer an analysis of atmospheric conditions over Western and Central Europe during the case study period, May and June 2013. Section 0 provides a summary for the days with the precipitation over 10 mm in May and short summary on overall weather conditions in June. A detailed summary on the first days in June in the territory of the Czech

Republic is given in Sect. 0.

### 3.1 Weather analysis, May 2013

Figure 2 presents the daily accumulated precipitation (24-hour) from May 1 to June 30, 2013. In May 2013, the precipitation was over 1 mm for 15 days from which, in 6 days, it reached 10 mm or more. In 4 consecutive days in June the precipitation was over 5 mm with a maximum of precipitation on June 2, 37 mm. In the period June 21–27 2013, there are 7 consecutive

days with 58 mm accumulated precipitation of rain. On May 5, the precipitation was associated with an upper level trough and weak gradients at 500 hPa. This cyclonic field over 500 hPa favored the development of convective clouds and the precipitation during May (a spring month) when the atmospheric dynamic increased. On May 11, a well-developed upper level trough was passing over Germany, Austria and the Czech Republic with intensification of upper level gradients at 500 hPa. An upper level cold front moving from West to East was associated with the trough. The 850 hPa temperature was 10º

C before the passage of the front while dropped to about 4º C after the passage. On May 20, Germany, Austria and the Czech Republic were under influence of the frontal part of a large upper level cyclone (500 hPa) and a developing surface cyclone moving into North-North East direction. The cold advection from West was detected at 850 hPa over the Czech Republic with the temperature dropping from 14º to 3º C. After May 23, a series of Atlantic cyclones approached Europe and passed from West to Central and then to North-East Europe. A well-developed upper level ridge (500 hPa) was seen over Western

Russia blocking the cold air mass associated with the Atlantic cyclones, see Figure 3. On May 23, the first sign of extension of the upper level ridge towards Baltic Region and North Scandinavia was observed (not shown). On May 31–31, the upper level Atlantic cyclone was blocked over Western Europe by an upper level ridge located to the East and an Atlantic anticyclone to the West. The center of the cyclone passed slowly towards Mediterranean Sea bringing humid warm air to Central Europe.

On June 2, 37 mm of precipitation were recorded in Prague. Figure 4 (left plot) shows a cloud band covering East Germany, West Poland, Austria and the Czech Republic which is collocated to the large scale precipitation (right plot).

### 3.2 Extreme precipitation events in the Czech Republic, June 2013

In June 2013, the Czech Republic was affected by three flood events. The extreme heavy precipitation event of June 1–3, 2013 was the most severe one with respect to the life lost and damage. It was followed by two other important precipitation

periods, from June 9 to June 11 and, finally, from June 23 to June 26, however with a smaller extent of the concerned area



and ensuing problems. Contrary to the first precipitation, the two smaller ones were rather well forecasted and therefore not further developed and studied in detail.

The first extreme precipitation was a consequence of a baroclinic instability developing over the Central European region, with a weak westward propagation and a slow decay. Its activity was maintained due to heat and moisture pumped from the Mediterranean Sea. Such situations are difficult to forecast in general, especially regarding the spatial distribution and quantity of precipitation.

Figure 5 shows the 24h precipitation amounts observed from June 1 (6h UTC) to June 2 (6h UTC), then from June 2 (6h UTC) to June 3 (6h UTC) and finally their 48h sum. The figure combines data from the professional meteorological stations and climate stations (altogether 760 stations), all with assured quality, and thus gives well the feeling of the episode at a glance. Firstly, a heavy precipitation is visible for the mountain regions (North-West, North, but also northern slopes of the Šumava mountains in the South-West). Secondly, there was a narrow belt of extreme precipitation crossing the country from South-West to North-East, affecting central Bohemia and the capital of Prague. The belt was a result of a chain-like effect existing for about 20 hours. From the point of view of the hydrological response, this was likely the most devilish combination of the timing, location and precipitation amounts which could happen, since peaks of water discharge from several smaller catchments met at the same time together in Vltava and Elbe main rivers in the central Bohemia. All local area and global numerical weather models had difficulties to forecast such key spatial distribution of rain and its amplitude mainly of the narrow belt. All models underestimated mainly the lowland amounts of precipitation and, another problem was recognized as the westerly shift of the core of the precipitation activity even in short forecast ranges.

In short, this event had a low predictability, characterized by a strong sensitivity to the initial conditions of the forecast. Therefore it is useful to cover this extreme event by the GNSS benchmark, which may in return provide better observation data for experiments to be carried out by the meteorological community.

## 4 Benchmark data set

Information about the Benchmark design including selected area and time period was given in Section 2. This section provides more detailed descriptions of data types collected for the campaign. The Benchmark contains data from these sources: a) GNSS observations, b) synoptic meteorological observations, c) NWM data and products, d) radiosonde observations, e) WVR observations, f) radar images and some other auxiliary data, all described in the following sub-sections. A spatial distribution of all stations collecting the various types of data and covered areas are shown in Figure 6.

### 4.1 GNSS data

Observations from 430 GNSS reference stations were collected in total from which 247 sites belong to the 'core' Benchmark domain. GNSS sites were organized in seven clusters on the basis of the country and its geographical part and labeled accordingly: 1) AT0, CZ0, DE0, DE1 and PL0 for the 'core' domain and 2) PL1, DE2, DE3 and DE4 for the 'extended'



domain, see Figure 1 and Figure 6. Additional 13 GNSS stations from the EUREF Permanent Network, EPN (Bruyninx et al., 2004) were selected as reference being outside of the Benchmark domain. These are necessary when processing double-difference observations for a reliable accessing of geodetic reference frame and for resolving tropospheric parameters in absolute sense (Duan et al., 1996). The EPN stations, displayed as black points in Figure 7, were collected into a special

cluster (EU0). Table 1 summarizes basic characteristics of the GNSS data.

All GNSS data files are provided in RINEX format and 30s sampling interval. A common distance between two stations is about 50–70 km as primarily designed for real-time kinematics positioning applications (e.g. Wübbena et al., 2005). From the total number of GNSS sites, 4 observed GPS, GLONASS and GALILEO satellites, 356 observed GPS and GLONASS satellites, and remaining 70 stations were equipped with GPS receivers only. Station metadata files were completed and

checked carefully. A qualitative and quantitative control and a standard positioning were performed using G-Nut/Anubis software (Václavovic and Douša, 2015) for all 23865 files of the campaign network and the 8-week period. Data characteristics were summarized in special files including results of code multipath estimation as well as phase cycle slips and clock jumps detection. Overall lists were prepared for coordinates and availability of GNSS constellations, signals and frequencies for all the sites.

Besides multiple metadata correction and station name consolidation, in total 15 sites had to be finally rejected from the data set because of the data issues: a) two sites with too many missing single- or dual-frequency observations, b) two sites displaced during the campaign period, c) six sites with many phase cycle slips detected, d) four sites with missing information about their instrumentation, and e) one site for a large coordinate repeatability from the reference solution processing.

**4.2 E-GVAP operational GNSS products**

Operational near real-time tropospheric solutions provided by 14 analysis centres (and 29 solutions) were collected for the campaign. These products contributed routinely to the E-GVAP (http://egvap.dmi.dk) fulfilling the requirements of the near real-time GNSS products as defined in TOUGH (2004). The products were provided every hour with a maximum latency of 90 min after the first processed observation. The solution thus relayed on the predicted part of the IGS ultra-rapid orbit

products (Springer and Hugentobler, 2001). Due to the lack of precise satellite clock corrections, the majority of solutions used the so-called network approach when processing double-differenced GNSS observations.

The products are stored in the COST-716 format with a temporal resolution from 5 to 60 minutes providing GPS zenith total delays for all available permanent stations in Europe. The products are included in the campaign mainly for the evaluation and inter-comparison purposes, but also for potential meso-scale model assimilation developments and studies.

**4.3 Synoptic data**

Meteorological measurements from 610 synoptic stations were collected. For all stations at least atmospheric air pressure, air temperature and relative humidity observations are available in the sampling interval from 10 to 60 minutes. Table 2





summarizes all information about meteorological parameters at synoptic stations collected for the Benchmark campaign. Original data were provided in various formats which were, additionally, converted into a single unified plain text format.

## 4.4 NWM data and products

NWM 3D data fields from the Czech Hydrometeorological Institute's (CHMI) local area model ALADIN-CZ were extracted
in GRIB format. The invariant NWM orography was included in a single static file, while meteorological parameters important for the derivation of GNSS-specific signal delays due to the troposphere were split into two epoch-wise specific data files. The former contains necessary parameters of the most commonly adopted model for the derivation of hydrostatic and non-hydrostatic refractivity to calculate corresponding GNSS signal path delays. The latter data type supports an extended model for calculating the effect on signal due to so called hydrometeors (e.g. ice and liquid water) which can be in
most cases neglected. All parameters are summarized in Table 3.

Characteristics of the CHMI's model setup are the following:

- Horizontal resolution: $4.7 \times 4.7$ km
- Vertical resolution: 87 model levels
- Time of analysis: 00, 06, 12, 18 UTC
- Forecast ranges: 00, 01, 02, 03, 04, 05, 06 hours
- Coordinates: non-rotated Lambert projection according to CHMI specification

The local area model domain is shown in Figure 7 as the shaded region. Additionally, GNSS stations of the campaign clusters are shown in coloured points and the EUREF Permanent GNSS stations in black points.

## 4.5 Radiosonde data

Data from two different sources were collected providing radio sounding profiles with full and reduced resolutions. Measurements with high resolution are available from two sites in the Czech Republic – Prague-Libuš and Prostějov, both provided by the Czech Hydrometeorological Institute. The first one (Vaisala RS92-SGP) launched at 00, 06, 12 and 18 UTC with 5s interval of measurement, and the second one (Vaisala RS80) at 00 and 12 UTC with 2s interval. Altogether 278 files were provided for the Benchmark time period.

Radiosonde data with reduced vertical resolution from 19 European stations were collected as provided by E-GVAP to the geodetic community based on the EUMETNET – EUREF MoU (Pottiaux et al., 2009).

## 4.6 Water Vapour Radiometer data

Measurements from Water Vapour Radiometers (WVR) were acquired for the Benchmark directly from the operators. Both instruments are situated in Germany. The first, HATPRO from Radiometer Physics, is operated by GeoForschungsZentrum
in Potsdam (POTS) 30 km south-westward from Berlin. The second, 12-channel MW-Profiler Radiometrics TP/WVP-3000,



is operated by Deutscher Wetterdienst (DWD) at the Lindenberg meteorological observatory (LDBG) located approximately 100 km eastward from Berlin. These two instruments were added mainly for the assessment of GNSS tropospheric gradients and slant delays. Therefore the Benchmark 'core' domain was extended northward up to Berlin.

The POTS WVR provided in the Benchmark period measurements of Integrated Water Vapour (IWV) and liquid water in the GPS satellites tracking mode (slant observations) as well as in the zenith direction. Due to switching between the two observing modes (zenith and satellite tracking) the temporal resolution is not uniform. In zenith mode the temporal resolution is about 30 seconds periodically interrupted by gaps of several minutes for satellite tracking. The LDBG WVR observed IWV values and liquid water in the zenith direction only and measurements are available in 10 minutes interval. Unfortunately, data from two additional radiometers within the Benchmark domain, GOPE and WTZR, were not available during the period of the campaign due to instrument malfunctioning.

### 4.7 Meteorological Radar images

Images of combined observations from two meteorological radars (Skalky, Brdy) located in the Czech Republic and operated by the Czech Hydrometeorological Institute were provided for informative purpose. Both instruments were non-polarimetric C-band Doppler radars. The Radar at Skalky was type Gematronik 360AC installed in 1995; receiver, software and transmitter modulator switch were upgraded in 2006. The radar in Brdy (EEC DWSR-2501C) was installed in 1999; receiver and software were upgraded in 2007.

The radar observations represent maximum reflectivity fields with side projections in horizontal resolution of 1x1 km and 30 minutes time interval. The area effectively covered by those two radars includes the territory of the Czech Republic and areas of approximately 100 km outside the Czech state boundary. Important areas of interest, the Danube and Elbe river basins affected by strong floods during the Benchmark time period, are thus within the combined radar images.

### 5 Initial analysis and reference products

The Benchmark campaign aimed at supporting an effective collaboration in the GNSS4SWEC project using a common, well-prepared and documented data set as described above. Various additional products and models are needed for the GNSS data processing, such as precise orbits and clocks, earth rotation parameters, transmitter and receiver antenna phase centre offsets and variation models, differential code biases, ionosphere maps and loading corrections. These are however available from the existing services such as the International GNSS Service, IGS (http://www.igs.org), the International Earth Rotation and Reference Systems Services, IERS (http://www.iers.org) and others and therefore were mostly not included in the data set itself.

After the data preparation, quality checking and cleaning, several other procedures were performed on all GNSS and NWM data sources. The first procedure was the GNSS post-processing for a final data and metadata consistency check and for generating reference coordinates and tropospheric parameters. The second procedure was dedicated to the purpose of GNSS



tropospheric product comparisons and conversions and for providing additional supplementary products for GNSS data processing such as a priori zenith tropospheric delays, horizontal tropospheric gradients and mapping function coefficients. All the complex solutions completed an overall assessment of all available data and reference product quality. They also provided a first insight into variations of parameters and atmospheric conditions. Altogether, these are helpful for more

detailed planning of future Benchmark-related activities, providing an initial feedback in development of advanced products and enabling a focus on specific time and space domains within different topics of interest.

## 5.1 Reference tropospheric products

The first reference tropospheric product was generated at the Geodetic Observatory Pecný (GOP) using the Bernese GNSS Software V5.2 (Dach et al., 2007) with the network processing approach using double-differenced GNSS observations. The

strategy for daily solutions was consistent with the GOP contribution to the EUREF Repro2 campaign. It included the state-of-the-art models approved for a high-accurate analysis for the European reference frame maintenance. The use of precise orbit products from the CODE Repro2 solution (Dach et al., 2014) guaranteed a full consistency of all implemented models on the provider and user side. The models were compliant with the IERS Conventions (2010). Characteristics of the campaign data processing for generating the reference GOP tropospheric product are summarized in Table 4.

The tropospheric parameters – zenith total delays and horizontal gradients – together with reference coordinates and other metadata, were stored in the TRO-SINEX format aimed to be updated within the GNSS4SWEC project for the dissemination of advanced tropospheric products. Mean coordinate repeatability from the processing of all stations during the 56-day period reached 5, 2 and 6 mm for X, Y and Z component, respectively. A single station was rejected from the campaign due to the exceeding repeatability: 13, 13 and 14 mm for X, Y and Z, respectively. The resulted coordinates were provided as

reference, for example for defining positions of stations for calculating meteorological and tropospheric parameters from NWM as described in the next section.

Since the campaign network was too large for processing in a single run, two solutions were generated independently for 'core' and 'extended' domains. Both solutions applied the same strategy and common 13 EUREF reference stations for the geodetic datum definition. Comparisons of individually estimated coordinates and tropospheric parameters from these

reference stations provided an initial quality-check between both processings. Mean differences of reference station coordinates achieved sub-millimeter level; root mean squares (RMS) of zenith total delay differences were below 1 mm for 12 stations and only a single station (JOZE) resulted with RMS of 1.5 mm. From these investigations, a high consistency of tropospheric parameters is thus considered between 'core' and 'extended' reference solutions.

The second reference tropospheric product completed recently at the German Research Centre for Geosciences (GFZ) was

generated by the GFZ EPOS software (Gendt et al., 2004; Ge et al., 2006) using undifferenced GNSS observations and PPP approach. For daily solutions with PPP we needed precise satellite orbits and clocks, which were estimated separately using the network processing approach with about 100 globally distributes IGS sites. The use of precise orbit and clock products from the GFZ solution guaranteed a full consistency of the GFZ tropospheric products. The models were compliant with the




IERS Conventions (2010). Characteristics of the campaign data processing for generating the reference GFZ tropospheric product are also summarized in Table 4.

**5.2 NWM-derived tropospheric parameters**

Parameters of the tropospheric models and corrections were derived from two different global numerical weather models – the ERA-Interim (Dee et al., 2011) from the European Centre for Medium-Range Weather Forecasts (ECMWF), and the Global Forecast System, GFS of the National Centres for Environmental Prediction (NCEP) available at http://www.ftp.ncep.noaa.gov/data/nccf/com/gfs/prod/. The ERA-Interim is a re-analysis product available every 6 hours (00, 06, 12, 18 UTC) with a horizontal resolution of 1×1 degree and 60 vertical model levels. The NCEP's GFS analyses are available every 6 hours (valid at 00, 06, 12, 18 UTC) with a horizontal resolution of 1×1 degree on 26 pressure levels. The processing of the Benchmark period was completed with the 3-hour forecasts (valid at 03, 09, 15 and 21 UTC). Both NWM data sets were processed with the German Research Centre for Geosciences's (GFZ) direct numerical simulation (DNS) tool (Zus et al., 2014) in order to derive the following parameters, see the GNSS tropospheric model in Appendix A:

- Zenith hydrostatic and zenith wet delays: ZHD and ZWD, respectively.
- Horizontal (1st, 2nd order) tropospheric gradients: $G_N$, $G_E$, $G_{NN}$, $G_{EN}$, $G_{EE}$.
- Coefficients of hydrostatic ($a_h$, $b_h$, $c_h$) and wet ($a_w$, $b_w$, $c_w$) mapping functions.

Additionally, data from the two global numerical weather models (ECMWF's ERA-Interim and NCEP's GFS) and one regional model (ALADIN-CZ) were processed with the G-Nut/Shu software developed at the Geodetic Observatory Pecný, GOP (Douša and Eliaš, 2014) designed for developing a GNSS tropospheric correction model using NWM data. The following tropospheric and meteorological parameters were calculated for all stations: zenith hydrostatic and wet delays, air pressure, partial water vapour pressure, mean temperature, temperature lapse rate, water vapour pressure and zenith wet delay exponential decay rates. The last three are generated for the vertical scaling of other model parameters. Horizontal approximations from NWM grid points to GNSS stations were performed using bilinear interpolation followed by vertical parameter scaling as described in Douša and Eliaš (2014).

All abovementioned NWM data processing were performed for all GNSS sites of the Benchmark using mean station positions from the GNSS reference solution. The zenith total delays derived from the GNSS reference solution and from various NWM data analyses were compared in the GOP-TropDB evaluating system (Györi and Douša, 2015). Table 5 summarizes comparison results of the GNSS reference tropospheric zenith total delays with those derived from NWMs. Mean statistics over all 430 sites demonstrated that high-resolution ALADIN-CZ model, although being predicted up to 6 hours, outperformed significantly both global re-analysis models in the Benchmark domain and period. A similar performance was observed for NWM's ZTDs compared to GNSS reference solutions using Bernese and EPOS-8 software and different processing strategy. It is interesting to note that the GOP solution is more consistent with the ERA-Interim reanalysis while the GFZ solution with GFS or ALADIN-CZ models. NWM-derived ZTDs using G-Nut/Shu and GFZ/DNS


software differs also at sub-millimetre level, which can be considered as a very good agreement taking into account a complexity of both software implementations designed for different purposes. Finally, a negative mean bias of about 5 mm was observed in NCEP's GFS product compared to both GNSS solutions.

Geographical maps of ZTD statistics over all stations and Benchmark period are shown in Figure 8. ZTD biases (left) and standard deviations (right) were calculated as ZTD differences between GOP's GNSS reference product and CHMI's ALADIN-CZ local area model (top), ECMWF's global model (middle) and NCEP's GFS global model (bottom). As already seen in k in the local area model performs better mainly in terms of precision represented by the standard deviation. Generally, a good homogeneity was observed from the statistical results. Exceptions exist mainly in relation to the orography which triggers larger differences between models particularly in mountain areas. The example is a southward part of the Benchmark network showing increased standard deviations and negative (wet) bias in the ECMWF's ERA-Interim ZTD performance. This shows that a complex terrain such as in the Alps is much better captured by the mesoscale model with up to 23 times better horizontal resolution. A prevailing negative bias in the NCEP's GFS ZTDs, as clearly seen in Table 1, varies among stations and doesn't seem to be related to the orography only. A similar bias was observed from the global statistics of GFS compared to GNSS and it has not been explained yet.

### 5.3 GNSS and NWM tropospheric wet delay maps

Since for the GOP reference tropospheric product the VMF1 mapping function and the corresponding a priori zenith hydrostatic component (Boehm et al., 2006) was applied, we assume that the estimated corrections represent the zenith wet delay. It could be thus visualized as zenith wet delays directly or converted into the integrated water vapour (IWV) using mean temperature parameters calculated from the G-Nut/Shu NWM data processing.

Zenith wet delay maps were created from GNSS processing results and three different NWM models (ALADIN-CZ, ERA-INTERIM, GFS) for the first comparisons. Figure 9 shows an example of ZWD maps generated from GNSS and NWM models results for May 31, 18:00 UTC. This time period was shortly preceding the flood event described in Section 0. Maps from different sources show a good agreement in terms of ZWD distributions. Significant differences however occur around the narrow belt of high ZWD values, representing an increased amount of water vapour, stretched from the northern Poland to the southwest direction across whole Germany. The same situation is shown in Figure 10 plotting tropospheric linear horizontal gradients discussed in the next section.

### 5.4 Comparison of horizontal gradients from GNSS and NWM

Horizontal tropospheric gradients are supposed to represent the effect of the first-order asymmetry in the troposphere and thus providing additional information to ZTDs or ZWDs. Zero a priori gradients were introduced in the GOP or GFZ reference GNSS analyses and thus all solutions are considered as independent. In this paper we aim at inter-comparing gradients only to stimulate future work with detailed focus on a) developing a fast production of high-resolution horizontal gradients from GNSS dense networks, b) studying their potential exploitation in severe weather event monitoring or



nowcasting, and c) optimizing tropospheric gradient modelling in GNSS analyses. These goals will be studied within future Benchmark activities, in particular to study gradients during quiet and severe weather conditions while taking a full benefit of the Benchmark dense GNSS network.

First tropospheric gradient maps and animations were prepared using a linear interpolation to an hourly resolution. Because

horizontal gradients can change quickly during severe weather situations, the interpolation will be replaced with an hourly gradient update rate when estimated from local area models (e.g. ALADIN-CZ) or even higher with GNSS processing performed in (near) real-time. The animation of tropospheric gradients estimated from NWM and GNSS data during the 8-week period demonstrated a very good agreement among compared products in terms of gradient directions and, usually, their magnitudes. A typical example is shown in Figure 10 with gradients estimated with two GNSS software packages and

strategies (upper plots) and two NWM global models (bottom plots) on May 31 2013, 18:00 UTC. The gradients point from each station to the azimuth of the local maxima of tropospheric (wet) delay correction which usually corresponds to the increasing amount of water vapour in the troposphere. From the gradient map animations, we noticed interesting details which could be observed from GNSS solutions using such a dense station network. These are supposed to represent local asymmetries in a water vapour distribution in the atmosphere.

In some cases, the gradient maps derived from GNSS and NWM show discrepancies in terms of the magnitude of gradient values while keeping consistency in directions. Usually, GNSS gradients tend to be significantly higher than NWM ones as clearly visible in Figure 10. An overall structure of gradient heading remains very similar among products while it deviates in specific areas only. However, we also observed clear GNSS gradient patterns which were fully missing in NWM gradient maps. In summary, more detailed patterns were observed in GNSS maps while smoothed, or even missing, in NWM maps if

an asymmetry occurs in the troposphere. The lower magnitudes in NWM are supposed to be partly due to a limited spatial resolution of global NWM models. Further investigation will be particularly focused on local area models. Anyway, in this context GNSS maps indicated a potential of added value in terms of information about the structure of the troposphere, in particular of the water vapour content. Finally, it should be noted that GNSS-derived gradients demonstrated a nice ability for producing a homogeneous tropospheric gradient field revealing details in the actual state of the troposphere. In this paper,

we showed only a situation with large horizontal gradients. However, it seems that even smaller gradients provide valuable information about the water vapour horizontal distribution in the atmosphere.

Finally, gradients from all GNSS Benchmark stations and the whole period were evaluated in GOP-TropDB. Table 6 shows a comparison of North-South (NS) and East-West (EW) tropospheric gradients from the two global NWM data sets and two GNSS post-processing solutions. Almost negligible mean biases were observed and a precision of 0.5 mm for NS and EW

gradients in all products was achieved (see Appendix A for gradient representation). Currently, we could not derive tropospheric gradients from a high-resolution model such as CHMI's ALADIN-CZ without a modification of the ray-tracing software, however, this is planned in the next phase of the Benchmark-related activity. In this context, the effect of hydrometeors, see Appendix B, will also be studied since they were neglected in the current NWM data processing.



## 6 Conclusion

We summarized motivations, requirements and objectives of the Benchmark campaign organized within the COST Action ES1206 (GNSS4SWEC) and described a complex data set prepared for this campaign. The campaign plays a key role in the inter-disciplinary and a highly collaborative effort within the COST Action particularly aimed at enhancing GNSS tropospheric models and products for scientific and applied meteorology.

Temporal and spatial domains of the Benchmark were selected in Central Europe including northern part of the Alps and May–June 2013 when severe heavy rain falls and floods occurred in the Czech Republic and in the catchment area of the Danube river. The episodes occurring during this period were described in order to understand the situations related to extreme weather conditions on specific days and in specific areas. All campaign data and metadata were carefully collected, checked and cleaned. GNSS and NWM data were processed to obtain the reference tropospheric and coordinate solutions for all GNSS stations and to generate other supplementary models or products from NWM data for future GNSS analyses.

First analyses and comparisons of tropospheric parameters derived from GNSS and NWM datasets provided an insight to achievable quality of tropospheric products, their similarities and complementarities. ZTDs from reference GNSS solution compared to global ERA-Interim re-analysis resulted in an accuracy at the 10-millimeter level in terms of RMS (with a negligible overall bias), the comparison to global GFS forecast showed accuracy at the 12-millimeter level with the overall bias of -5 mm and, comparison to meso-scale ALADIN-CZ forecast resulted in an accuracy at the 8-milllimetre level with a negligible total bias.

Initial comparisons of horizontal tropospheric gradients retrieved from the dense GNSS network and global numerical weather models demonstrated a very good agreement in general, at the level of 0.5 mm. It has been shown that GNSS gradients are able to provide interesting information about detailed structure of the atmosphere, mainly related to the water vapour distribution. Certainly, these are of particular interest for numerical or non-numerical nowcasting applications and for monitoring severe weather events.

An initial study on contribution of hydrometeors to GNSS signal delays during severe weather is given in Appendix B, see Figure 11 and Figure **12**. The maximum GNSS zenith path delay caused of hydrometeors reached up to 17 mm during June 23, 2013. If liquid and ice water components are available in NWM output, we recommend considering them in the observation operator to minimize the cost function in the assimilation process or provide more accuracy product for GNSS positioning. In future, our attention will also focus on the impact of hydrometeors on gradient and slant delays simulations.

The results presented in this paper will be further used for more specific studies on defined topics and interests when using the Benchmark data set. Such studies are however out of the scope of this paper, which aimed at describing the Benchmark data set together with introducing and assessing reference products. Nevertheless, initial results already indicated a potential of advanced GNSS tropospheric products for meteorological applications and emphasized a synergy in GNSS and meteorological data and products. The Benchmark's careful design, data collection and, finally, preparation of reference and



auxiliary products is going to play a key role in further developing, assessing and exploiting advanced GNSS tropospheric products in meteorological applications within the GNSS4SWEC project.

**Acknowledgement**

The Benchmark campaign has been organized within the COST ES1206 Action. The authors thank all the institutions that
provided data for the campaign: GNSS data from the Austrian network EPOSA by Österreichische Bundesbahnen Infrastruktur AG; GNSS data from SAPOS network in Germany by Zentrale Stelle SAPOS in Hannover; GNSS data from several networks in the Czech Republic - 1) CZEPOS by the Czech Land Survey Office, 2) Trimble VRS Now® by GEOTRONICS Praha, s.r.o. and 3) GEONAS and VESOG stations thanks to the project CzechGeo (LM2010008) operated by the Institute of Rock Structure and Mechanics of the Academy of Sciences of the Czech Republic and the Research
Institute of Geodesy, Topography and Cartography, respectively; GNSS data from Polish ASG-EUPOS network by the Head Office of Geodesy and Cartography in Poland; Synoptic data by Zentralanstalt für Meteorologie und Geodynamik, ZAMG (Austria), Deutscher Wetterdienst, DWD (Germany), Czech Hydrometeorological Institute, CHMI (the Czech Republic) and Polish Institute of Meteorology and Water Management (Poland). Finally, special thanks come to people assisting in collecting all the data and metadata, namely: Dr. Jan Řezníček (GNSS/CZEPOS), Dr. Uwe Feldmann-Westendorff and Dr.
Markus Ramatschi (GNSS/SAPOS), Dr. Petr Novák (RADAR/CHMI), Dr. Martin Motl (RAOBS/CHMI), Dr. Anna Valeriánová (SYNOP/CHMI), Dr. Pavla Skřivánková (CHMI), Dr. Roland Potthast (SYNOP/DWD), Dr. Jürgen Güldner (WVR-Lindenberg/DWD) and Dr. Stefan Heise (WVR-Potsdam/GFZ). J. Douša and M. Kačmařík acknowledge the support for the preparation of the Benchmark campaign from the Czech Ministry of Education, Youth and Sports (Project No. LD14102).

**Appendix A: GNSS tropospheric model**

The tropospheric delay $T$ is approximated as

$$T(e,a) = m_h(a_h,b_h,c_h,e)ZHD + m_w(a_w,b_w,c_w,e)ZWD + \\ m_g(e)\left[ G_N\cos(a) + G_E\sin(a) + G_{NN}\cos^2(a) + G_{NE}\cos(a)\sin(a) + G_{EE}\sin^2(a) \right] \quad (0.1)$$

where, $e$ and $a$ are elevation and azimuth angles for specific satellite, $ZHD$ and $ZWD$ are the so-called zenith hydrostatic and wet delays, respectively, $m_h$, $m_w$, $m_g$ are the so-called hydrostatic, wet and gradient mapping functions with specific
coefficients $a_h$, $b_h$, $c_h$ and $a_w$, $b_w$, $c_w$, respectively, and $G_N$, $G_E$, $G_{NN}$, $G_{NE}$, $G_{EE}$ are the components of the first order and the second-order horizontal tropospheric gradients.

The elevation angle dependency of the hydrostatic (non-hydrostatic) mapping function is based on the continued fraction form proposed by Marini (1972) and normalized by Herring (1992) to yield the unity at zenith



$$m(a,b,c,e) = \frac{1+a/(1+b/(1+c))}{\sin(e)+a/(\sin(e)+b/(\sin(e)+c))} \qquad (0.2)$$

The elevation angle dependency of the gradient mapping function is based on the form proposed by Chen and Herring (1997)

$$m_g(e) = \frac{1}{\sin(e)\tan(e)+c} \qquad (0.3)$$

with c = 0.003.

The model is designed in such a way, that only selected parameters (total delay in zenith direction and, optionally, linear horizontal gradients) need to be necessarily adjusted from the GNSS observations to achieve a sub-centimetre accuracy in positioning. However, by applying the same model on NWM data fields and using the DNS tool (Zus et al., 2014), all the tropospheric parameters $ZHD$, $ZWD$, $a_h$, $b_h$, $c_h$, $a_w$, $b_w$, $c_w$, $G_N$, $G_E$, $G_{NN}$, $G_{NE}$, $G_{EE}$ can be determined for specific positions of

GNSS reference stations defined by their coordinates. The underlying Mapping Factors (MFs) and Slant Factors (SFs), i.e. the ratios of slant and zenith delays, are ultra-rapid MFs and SFs (Zus et al., 2015a). From the NWM data the tropospheric parameters are estimated by least-square fitting. Note that this is done separately for the $a$, $b$ and $c$ coefficients of the mapping function and the gradient components. NWM-based parameters are then introduced in GNSS processing while ZTD only and, optionally, linear horizontal gradients, are re-adjusted from the GNSS observations.

**A. 1 Mapping function coefficients – a, b, c**

For each station 10 hydrostatic (non-hydrostatic) MFs are computed for elevation angles of 3, 5, 7, 10, 15, 20, 30, 50, 70, 90 degrees and the hydrostatic (non-hydrostatic) mapping function coefficients are determined by least-square fitting.

**A. 2 Horizontal tropospheric gradients**

The gradient components for each station are computed as follows; at first, 120 SFs and corresponding MFs are computed

(the elevation angles are 3, 5, 7, 10, 15, 20, 30, 50, 70, 90 degrees and the azimuth angles are 0, 30, 60, 90, 120, 150, 180, 210, 240, 270, 300, 330 degrees). Second, zenith delays are applied to obtain azimuth-dependent and azimuth-independent slant total delays. Third, the differences between azimuth-dependent and azimuth-independent slant total delays are computed. Finally, the gradient components are determined by least-square fitting. Note that an elevation angle dependent weighting, i.e., $1/cos^2(z)$ where $z$ denotes the zenith angle, is applied (Zus et al., 2015b).



**Appendix B: Functional relation between NWM meteorological parameters and GNSS tropospheric model**

The commonly used functional relation between GNSS tropospheric parameters in zenith and NWM meteorological parameters is the following:

$$ZHD = 10^{-6} \int_0^\infty N_h dz = 10^{-6} k_1 R_d \int_0^\infty \rho_m dz \,, \qquad (0.4)$$

$$ZWD = 10^{-6} \int_0^\infty N_v dz = 10^{-6} \int_0^\infty \left( k_2' \frac{e}{T} + k_3 \frac{e}{T^2} \right) dz \qquad (0.5)$$

where $N_h$, $N_v$ [-] are the hydrostatic and non-hydrostatic refractivities, respectively, $\rho_m$ [kg.m$^{-3}$] is the mass density of the moist air (i.e. the sum of densities of dry gases and water vapour), $R_d$ [J.K$^{-1}$.mol$^{-1}$] is the specific gas constant of dry air, $e$ [hPa] is the partial pressure of water vapour, $T$ [K] is the temperature, $z$ [m] is the geopotential height and, $k_1$, $k_2$, $k_3$ are empirical coefficients (Bevis et al., 1994).

Generally, using the above functional relation for ZHD and ZWD, the estimation of water vapour content from GNSS delay is really good, showing reliable results in comparison to other techniques (Van Malderen et al., 2014). However, during severe weather conditions, the GNSS signal is not only sensitive to the total density of the neutral atmosphere (so called ZHD) and to the specific additional contribution of water vapour (so called ZWD). Additional contribution and effect can take place from other particulates (Solheim et al., 1999), i.e. hydrometeors. More details about hydrometeors contributions in high resolution non-hydrostatic model were presented in Brenot et al. (2006). The additional contribution of hydrometeors to the ZTD (so called ZHMD) at the frequency of GNSS signal takes the following formulation:

$$ZHMD = 10^{-6} \int_0^\infty \left( N_{lw} + N_{ice} \right) dz = \int_0^\infty \left( 1.45 M_{lw} + 0.69 M_{ice} \right) dz \qquad (0.6)$$

where $M_{lw}$ is the mass content of liquid water hydrometeors (e.g. cloud water and rainwater) and $M_{ice}$ the mass content of icy hydrometeors (e.g. pristine ice, snow and graupel).





For the Benchmark campaign, the parameterization of the microphysics applied for CHMI's ALADIN-CZ model provides the mixing ratios of cloud water (liquid components) and pristine ice (solid water components). The impacts on ZTDs of the delays induced by the hydrometeors (cloud water and pristine ice) have been evaluated for the whole period of the Benchmark campaign. Figure 11 shows the temporal evolution of the ALADIN-CZ domain (4.7 km resolution) affected by ZHMD during June 2013. The surface of the domain simulated is about $10^6$ km² (latitude from 46°N to 56°N and longitude from 6°E to 20°E). The maximum values of ZHMD simulated for May 2013 do not reach more than 8 mm (not presented), however the figure presents maximum contribution of hydrometeors up to 17 mm during June 2013 (extreme weather on 23 June). During this event, characterized by the large scale convection and the path of a weather front from West to East, 30% of the domain (300000 km²) is affected by ZHMD > 0.003 m, 9% (90000 km²) by ZHMD > 0.006 m, 3% (30000 km²) by ZHMD > 0.009 m and 1% (10000 km²) by ZHMD > 0.012 m. The corresponding errors/overestimations of IWV (retrieved by GNSS technique) are respectively 0.5 kg/m², 1 kg/m², 1.5 kg/m² and 2 kg/m². Figure 12 presents ZHMD simulation on 23 June, 13:00 UTC. The surfaces affected by ZHMD for the 4 classes, as described in Figure 11, are respectively 183000 km² (17% of the domain), 77000 km² (7%), 23000 km² (2%) and 4000 km² (0.4%).

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



Table 1: Characteristics of Benchmark GNSS data set.

| Country / networks (clusters) | Height [m] min/max | GPS # sites | GPS+GLO # sites |
|---|---|---|---|
| Austria / EPOSA (AT0) | 168/2215 | 29 | 28 |
| Czech Rep / CZEPOS, VESOG, GEONAS, Trimble (CZ0) | 217/1375 | 73 | 45 |
| Germany / SAPOS (DE0, DE1, DE2, DE3, DE4) | 49/1828 | 282 | 281 |
| Poland / ASG-EUPOS (PL0, PL1) | 35/510 | 46 | 6 |



Table 2: Characteristics of synoptic data collected for the Benchmark campaign.

| Country | Number of stations | Available parameters | Time interval (minutes) |
|---|---|---|---|
| Austria (AT) | 280 | P, T, RH | 10 |
| Czech Republic (CZ) | 37 | P, T, RH, Rain/hour | 60 |
| Germany (DE) | 219 | P, T, Td, RH | 60 |
| Poland (PL) | 75 | P, T, RH | 60 |



Table 3: Content of the ALADIN-CZ NWM data.

| File name | Representation | Parameters |
|---|---|---|
| ALASMODL.YYYYMMDDHH.FC.grb | Epoch-wise | Atmospheric pressure [Pa] |
| | | Temperature [K] |
| | | Relative humidity [%] |
| ALASMODLW.YYYYMMDDHH.FC.grb | Epoch-wise | Atmospheric liquid water [kg/kg] |
| | | Atmospheric solid water [kg/kg] |
| SURFGEOPOTENTIEL.grb | Static | Surface geopotential [$m^2/s^2$] |



Table 4: Characteristics of GOP and GFZ reference tropospheric product solution with individual strategy indicated by corresponding acronym in the second column.

| Processing options | | Description |
| --- | --- | --- |
| Products | | Precise orbit and earth rotation parameters. |
| | GOP | CODE Repro2 products. |
| | GFZ | GFZ precise orbits and earth rotation parameters calculated using 100 global sites. |
| Observations | | Dual-frequency code and phase GPS observations from L1 and L2 carriers. |
| | GOP | Elevation cut-off angle 3 degree, elevation-dependent weighting $1/\cos^2$ (zenith), double-difference observations and with 3-minute sampling rate. |
| | GFZ | Elevation cut-off angle 7 degree, elevation-dependent weighting $1/2\cos(\text{zenith}>60\text{deg})$, undifferenced observations with 2.5-minute sampling rate. |
| Ref. frame | | IGb08 realization, core stations set as fiducial after a consistency checking. |
| | GOP | Coordinates estimated using a minimum constraint. |
| | GFZ | No constraint for coordinates since it is implicitly given in PPP by positions of satellites. |
| Antenna model | GOP | GOP: IGS08_1832 model (receiver and satellite phase centre offsets and variations). |
| | GFZ | GFZ: IGS08_1854 model (receiver and satellite phase centre offsets and variations). |
| Troposphere | | A priori zenith hydrostatic delay using VMF1 model and mapping function. |
| | GOP | Estimated ZTD corrections every hour using VMF1 wet mapping function; 5 m and 1 m for absolute and relative constraints, respectively. |
| | | Estimated horizontal NS and EW tropospheric gradients every 6 hours with no a priori tropospheric gradients and very loose absolute/relative constraints. |
| | GFZ | Estimated ZTD corrections every 15 minutes using VMF1 wet mapping function. |
| | | Estimated horizontal NS and EW tropospheric gradients every hour with no a priori tropospheric gradients and very loose absolute/relative constraints. |
| Ionosphere | | Eliminated using ionosphere-free linear combination with applying higher-order effects estimated using CODE global ionosphere product. |
| Loading effects | | Atmospheric tidal applied. |
| | | Hydrostatic loading not applied. |
| | | Ocean tidal loading applied (FES2004). |
| | GOP | Atmospheric non-tidal loading applied using the model from TU-Vienna. |
| Gravity | | EGM2008 model. |





Table 5: Comparison of zenith total delays from NWM and GNSS (mean values, 430 sites).

| NWM source (software) | Grid resolution | Analysis [hour] | Forecast [hour] | GNSS source (software) | Pairs # | Excl # | Bias [mm] | Sdev [mm] | RMS [mm] |
|---|---|---|---|---|---|---|---|---|---|
| ERA (Shu) | 1 deg | 6 | 0 | GOP (Bernese) | 224 | 2 | +0.0 | 9.6 | 10.0 |
| ERA (Shu) | 1 deg | 6 | 0 | GFZ (EPOS-8) | 224 | 3 | +0.3 | 9.7 | 10.0 |
| ERA (DNS) | 1 deg | 6 | 0 | GOP (Bernese) | 224 | 3 | -0.4 | 9.4 | 9.8 |
| ERA (DNS) | 1 deg | 6 | 0 | GFZ (EPOS-8) | 224 | 3 | -0.1 | 9.6 | 9.8 |
| GFS (DNS) | 1 deg | 6 | 3 | GOP (Bernese) | 224 | 7 | -4.9 | 11.0 | 12.0 |
| GFS (DNS) | 1 deg | 6 | 3 | GFZ (EPOS-8) | 223 | 7 | -4.5 | 10.9 | 11.8 |
| ALADIN (Shu) | 4.7 km | 6 | 0,1,2,3,4,5 | GOP (Bernese) | 1343 | 20 | +0.8 | 7.6 | 7.8 |
| ALADIN (Shu) | 4.7 km | 6 | 0,1,2,3,4,5 | GFZ (EPOS-8) | 1343 | 22 | +0.6 | 7.3 | 7.5 |



Table 6: Comparison of NS/EW tropospheric gradients from global NWM (DNS tool), GNSS/GOP (Bernese software) and GNSS/GFZ (EPOS-8 software); mean values (430 sites).

| NWM source | GNSS source | NS gradients | | | | EW gradients | | | |
|---|---|---|---|---|---|---|---|---|---|
| | | Pairs (excl) | Bias [mm] | Sdev [mm] | RMS [mm] | Pair (excl) | Bias [mm] | Sdev [mm] | RMS [mm] |
| ERA | GOP | 224 (4) | -0.02 | 0.41 | 0.42 | 224 (3) | -0.04 | 0.43 | 0.46 |
| ERA | GFZ | 224 (3) | +0.14 | 0.51 | 0.53 | 224 (3) | -0.08 | 0.49 | 0.50 |
| GFS | GOP | 224 (5) | -0.04 | 0.44 | 0.45 | 224 (4) | -0.05 | 0.46 | 0.50 |
| GFS | GFZ | 224 (3) | +0.13 | 0.54 | 0.56 | 224 (4) | -0.09 | 0.53 | 0.55 |





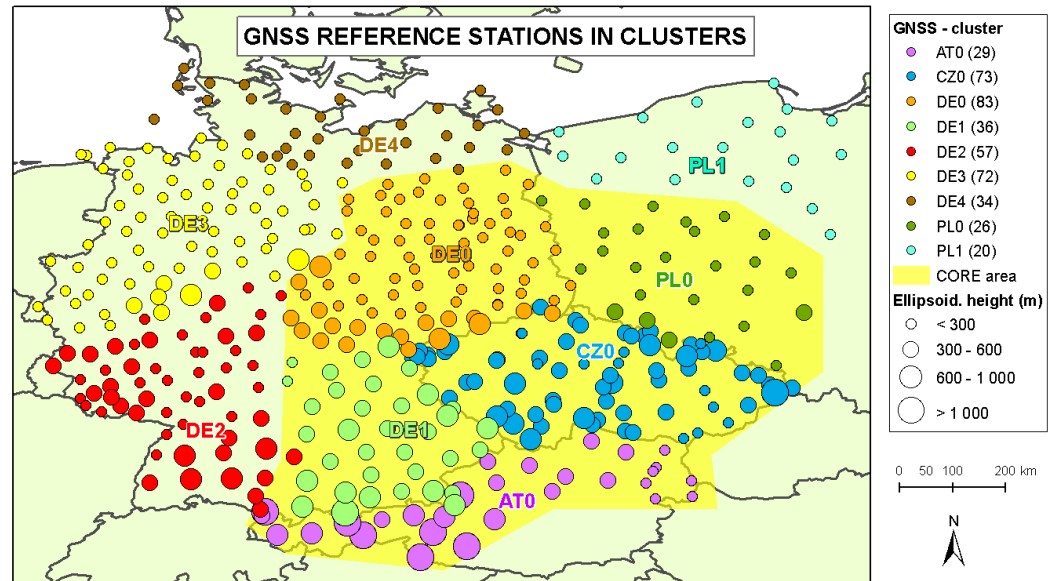

Figure 1. Benchmark 'core' (yellow area) and 'extended' domains depicted together with 9 clusters for GNSS stations (colour points). The size of the points indicates GNSS height above WGS84 ellipsoid.





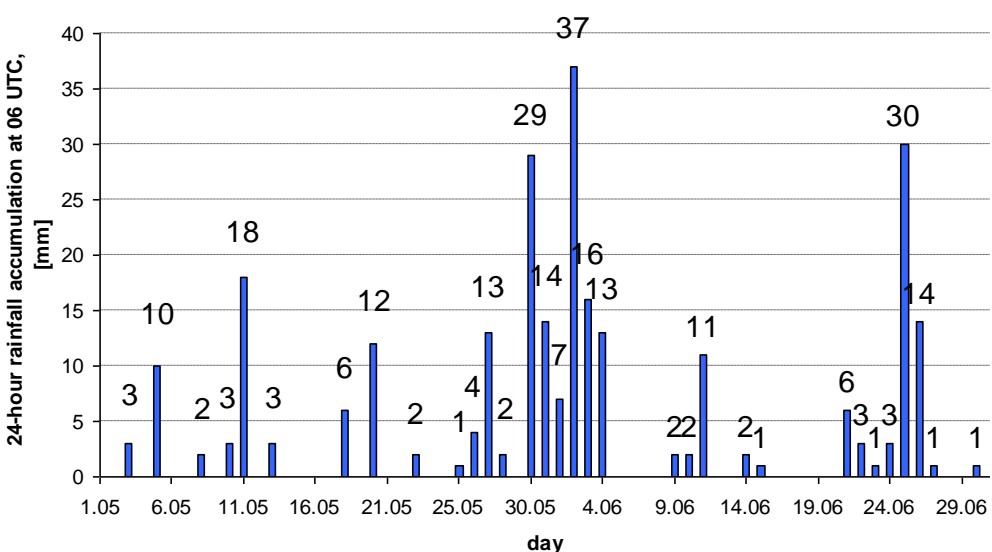

Figure 2. Daily precipitation from May 1 to June 4, 2013 at Prague-Ruzyne (11518) synoptic station.





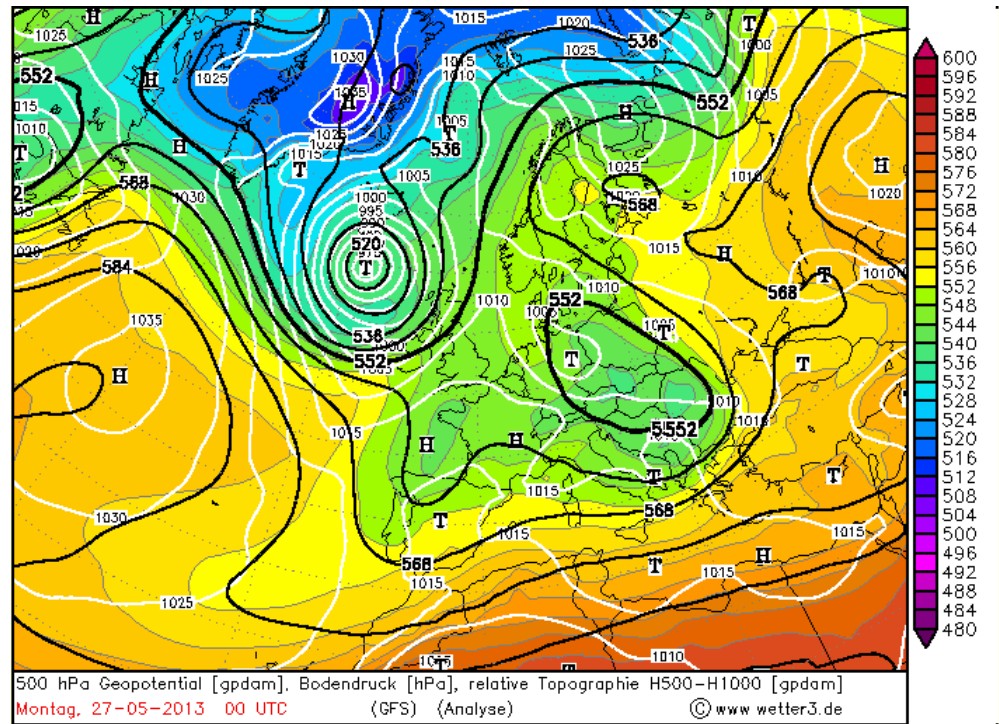

Figure 3. Thickness chart (in colour), geopotential height at 500 hPa (black line) and surface pressure (white line) on May 27, 2013 at 00 UTC.



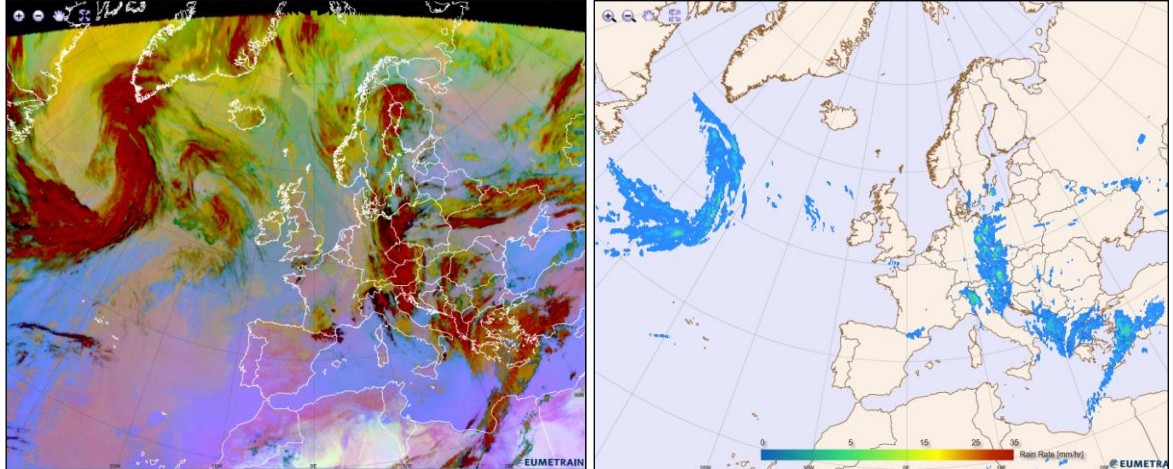

Figure 4. Weather analysis on June 2, 2013. Left plot shows cloud cover from Meteosat Second Generation (MSG) with red colour indicating thick clouds with ice particles. Right plot shows instantaneous rain rate product as MPE (Multi-sensor Precipitation Estimate) derived from the IR data of the geo-stationary EUMETSAT satellites by continuous re-calibration of the algorithm with rain-rate data from polar orbiting microwave sensors. Plots available from: http://www.eumetrain.org/eport/view.php?date=2013060200®ion=euro.





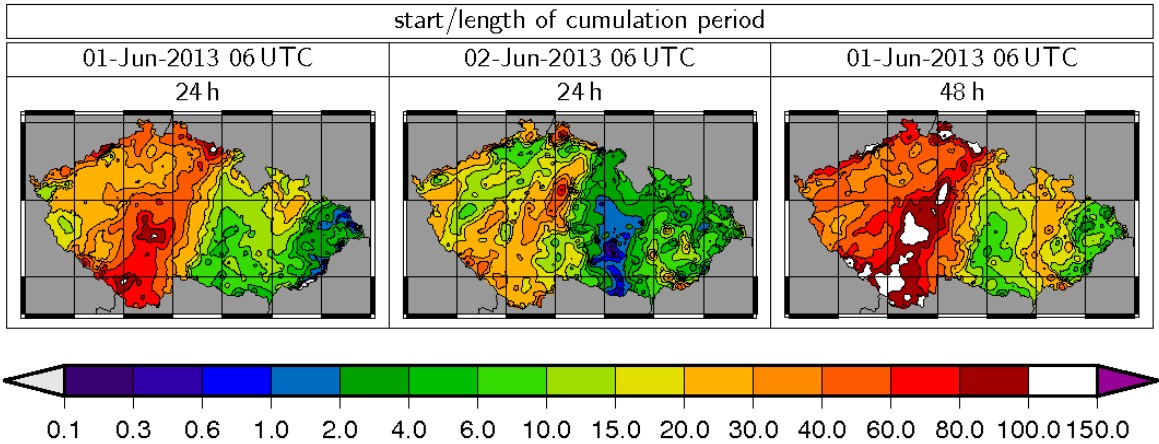

Figure 5. Precipitation amounts measured from 6h UTC June 1 to 6h UTC June 2 (left), from 6h UTC June 2 to 6h UTC June 3 (middle) and their sum (right).



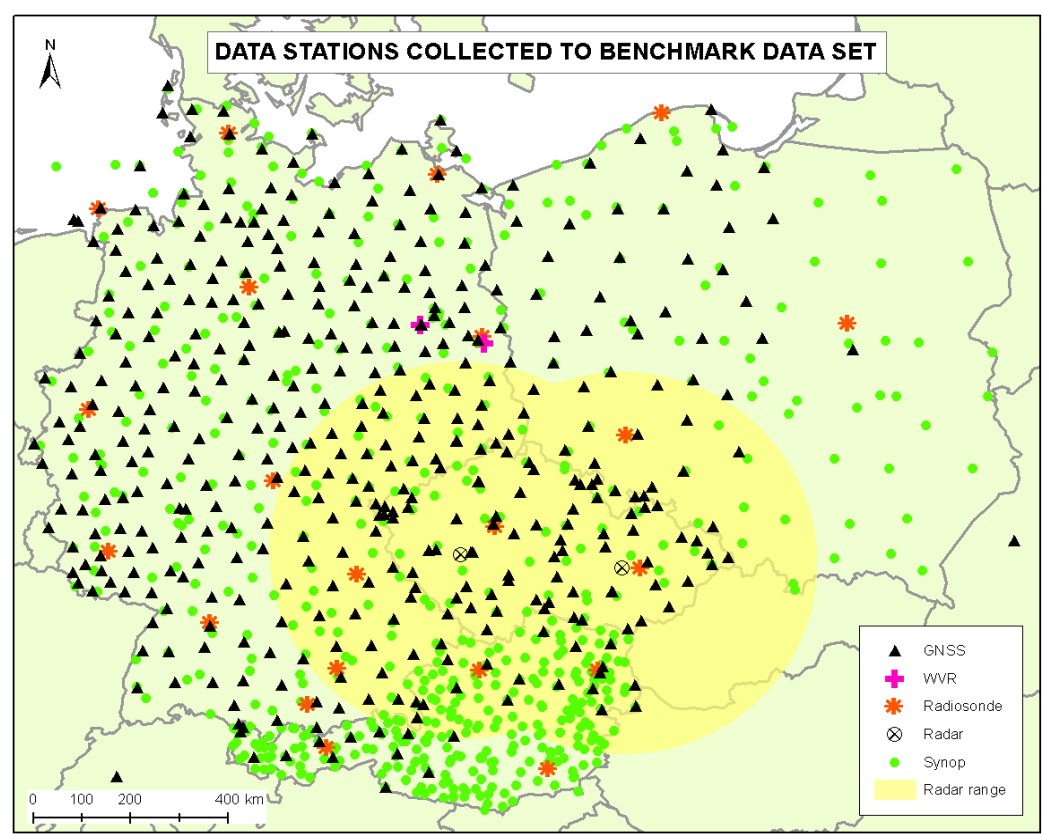

Figure 6. All type data stations collected within the Benchmark dataset.





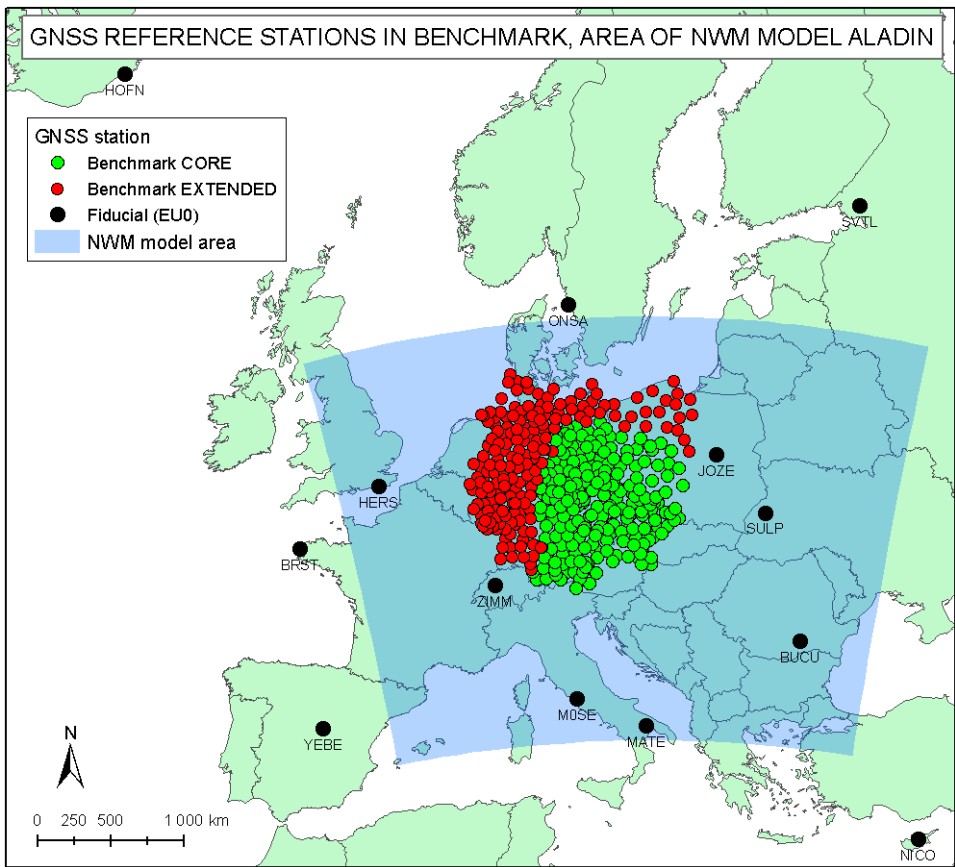

Figure 7. NWM ALADIN-CZ domain (shaded area) together with GNSS sites in Benchmark 'core' (green points) and 'extended' (red points) domains and fiducial sites (black points).





Figure 8. Maps of station-by-station comparisons of zenith total delays (ZTD) estimated from GOP's GNSS reference solution and two numerical weather models (GNSS-NWM) 1) CHMI's ALADIN-CZ (top), 2) ECMWF's ERA-Interim (middle) and 3) NCEP's GFS (bottom). ZTD biases in millimetres are shown in left plots and standard deviations in right plots.





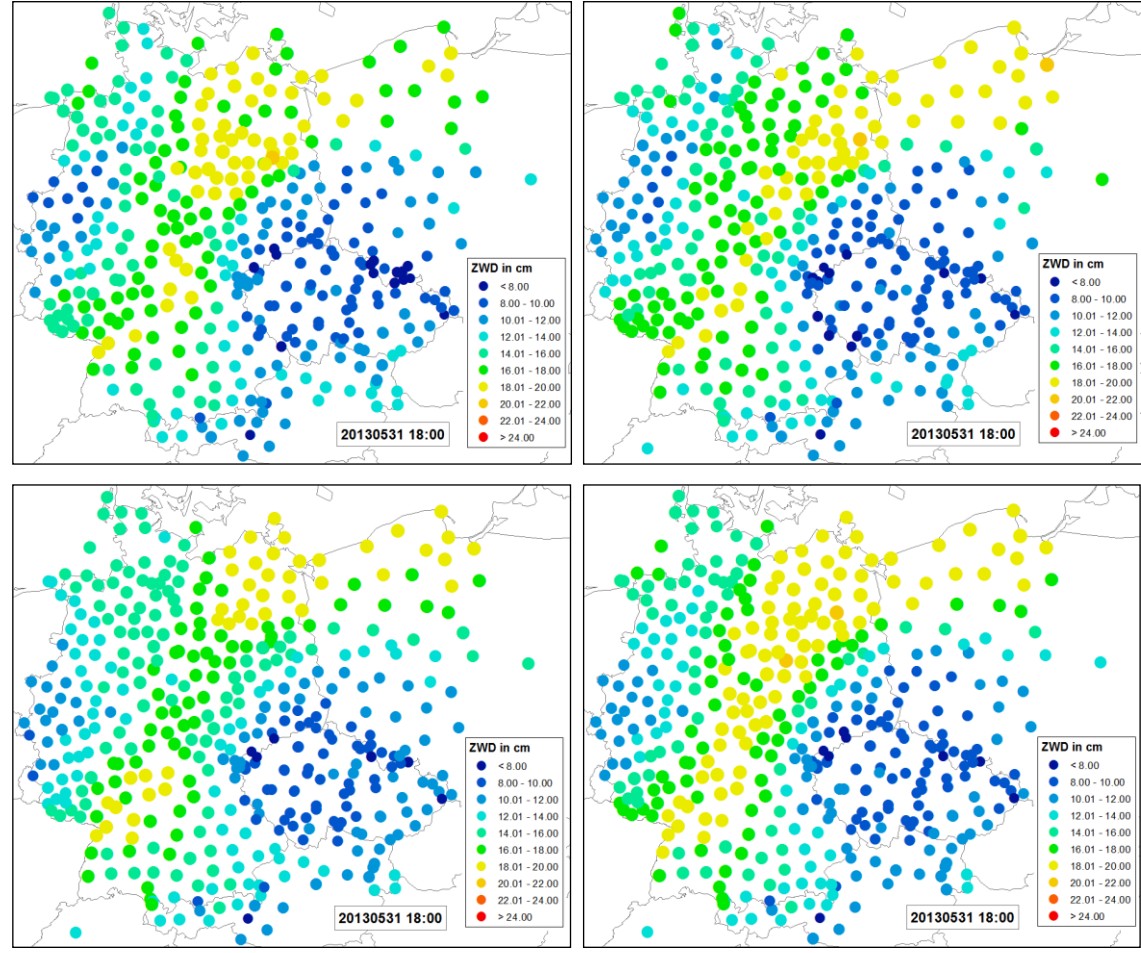

Figure 9. Fields of zenith wet delays estimated for all stations from GNSS processing (top left),
ALADIN-CZ model (top right), ERA-INTERIM model (bottom left) and GFS model (bottom right) at
May 31 2013 (18:00 UTC).





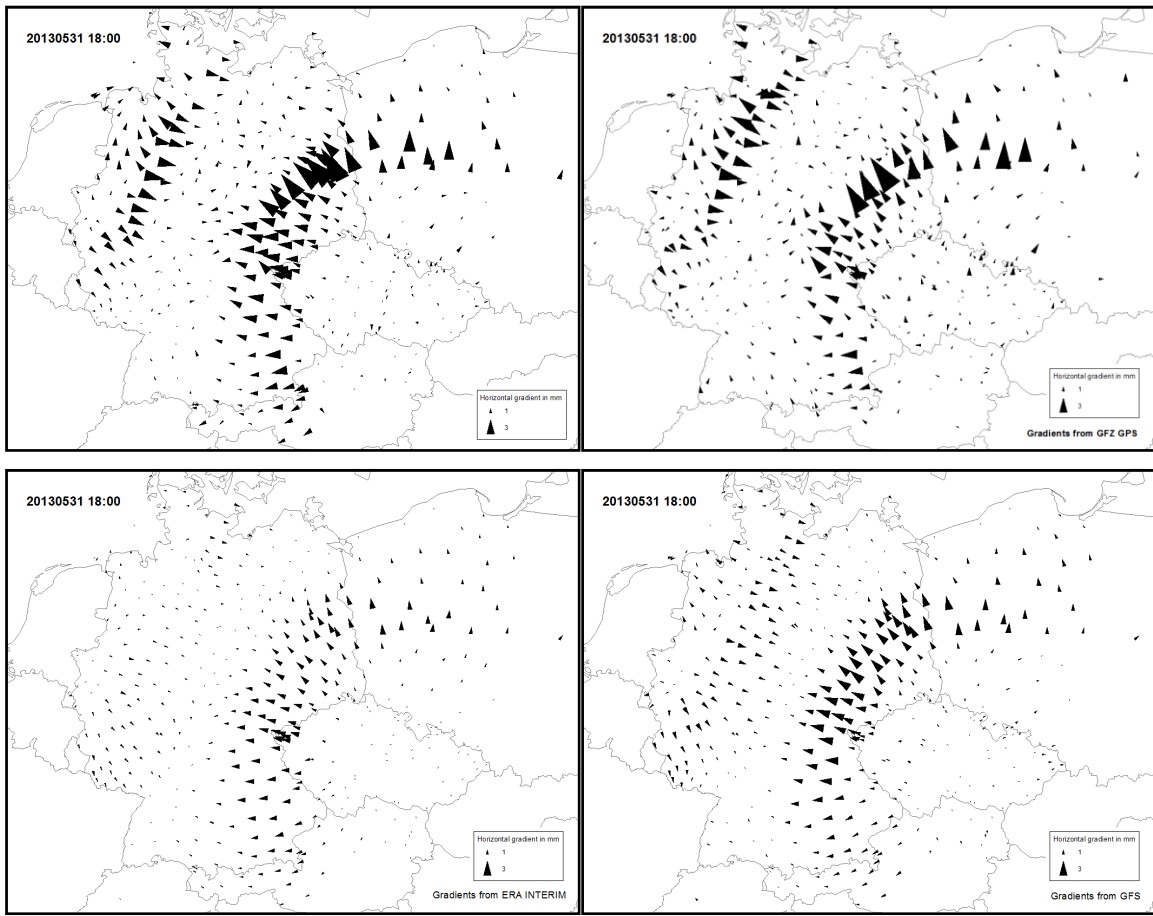

Figure 10. Tropospheric horizontal gradient maps from GNSS reference solution (top: GOP left and GFZ right) and from NWM global models (bottom: ERA-Interim left and GFS right) on May 31 2013 (18:00 UTC).





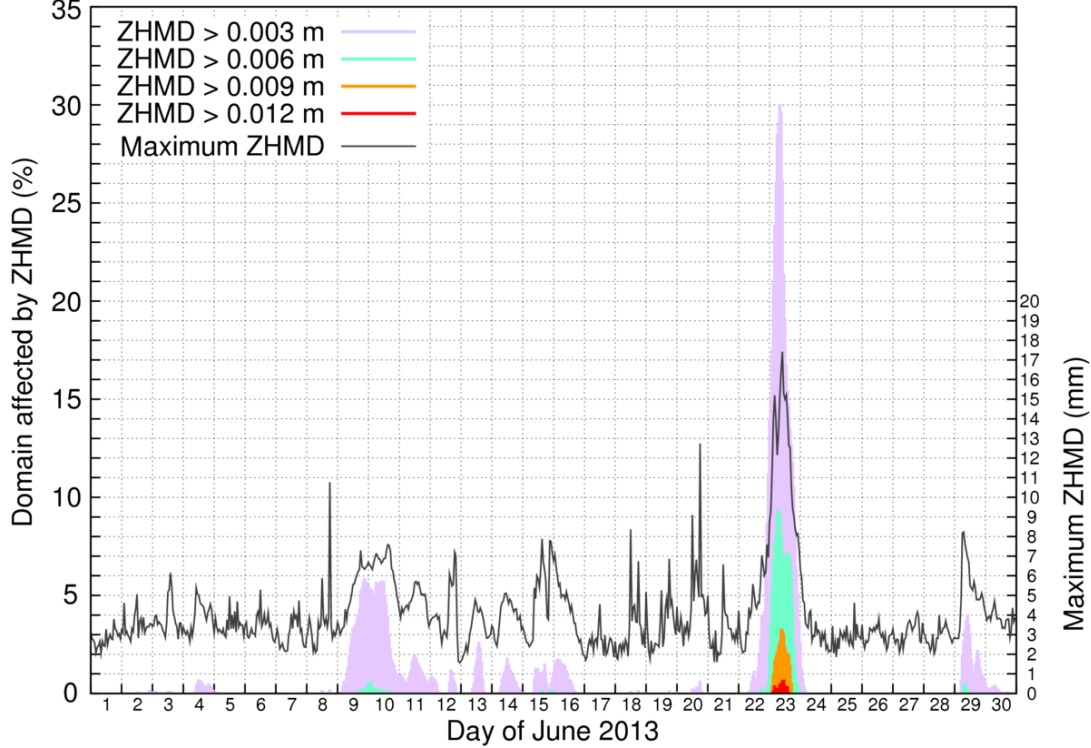

Figure 11. ZHMD contributions to zenith delays simulated with ALADIN-CZ model (June 2013) ranking in four classes (ZHMD > 0.003 m, ZHMD > 0.006 m, ZHMD > 0.009 m, and ZHMD > 0.012 m) shown respectively in purple, turquoise blue, orange and red. The maximum ZHMD (in mm) is shown with a dark grey line.



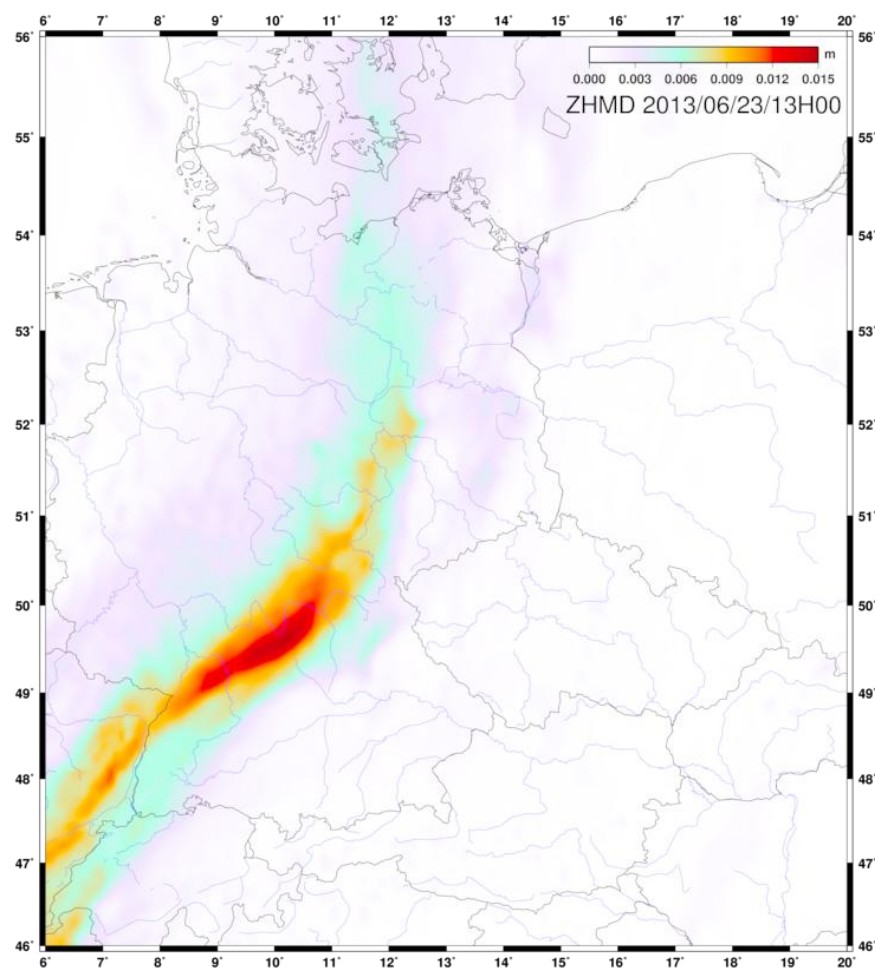

Figure 12. Image of ZHMD simulated with ALADIN-CZ on 23 June, 13H UTC.