# Peer review of "Benchmark campaign and case study episode in Central Europe for development and assessment of advanced GNSS tropospheric models and products"

_Atmospheric Measurement Techniques, 2015_

## Referee Comment (RC1) · Anonymous Referee #1 · 17 Mar 2016

This is a very fine overview article, providing a short overview of the GNSS4SWEC ES 1206 Cost action, an extensive, detailed overview of the selection and handling of data for the GNSS4SWEC benchmark campaign, and some first results from the inter comparisons of GNSS and NWP derived ZTDs and ZTD gradients.

It is clear the benchmark data have been carefully screened and processed, providing a very useful dataset for further studies. Among the most interesting results are that GOP and GFZ results agree well (network versus ppp), that higher resolution NWP seems ot agree better with the GNSS, and that GFS NWP is biased with respect to the

other solutions.

The the NWP community the finding of significantly larger gradients from the GNSS processing than found in NWP is very interesting, and potentially useful as an added data source. Likewise the finding that in certain situations the contribution from liquid water and ice is not negligible (estimated from NWP based calculations).

In my view the manuscript is almost ready for acceptance. I would recommend correct points 1 and 3 below before acceptance, while point 2 might be better suited by an online comment from the authors, given that this type of manuscript does not contain many details.

A few more detailed comments.

1 Many places in the text a reference to section 0 is given. Presumably that's the annex (judging from the numbering of the equations there), but the annex has no number.

2 A bias for GFS ZTD is found. There could be several reasons, some of which are related to how the numerical integrator for NWP ZTD is made, regarding both interpolation (different vertical resolutions) and extrapolation (top of the model).

3 Equation 0.6, for the effect of the hydro meteors, contains numbers on the right hand side, which seem to call for units. Is what is called "mass content" of hydro meteors another word for their mass density?

Speaking about units there is on the other hand no need to provide units for the density, pressure, temperature and gas constants further up page 19, as long as no values are given for the constants in equations 0.4 and 0.5.

The idea to include the hydro meteor contribution in the NWP cost function (if the NWP contains the relevant parameters) mentioned in the conclusion, is interesting. It reality is is very complicated, since they are normally not variables in the same sense as specific humidity of the NWP, and the NWP can be far of regarding their size. But pointing out that hydro meteors do sometimes contribute significantly to what appears

otherwise to be ZWD is important.

---

## Referee Comment (RC2) · Anonymous Referee #2 · 18 Mar 2016

Specifications and characteristics of the Benchmark campaign data set are described in detail. I believe the data set will contribute for both geodesy and meteorology in not only Europe but other countries and regions. However, there are not a few typos, insufficient explanations in body, tables, and figures. I would like authors to carefully polish the article. Followings are for reference.

About the chapter arrangement: (1). It would be nice to move "3 Case study episode in 2013" to after "4. Benchmark data set" and before "5. Initial analysis and reference products" (2)Effects of hydrometeor described in the latter half of the "6. Conclusion"

[Figure]

and latter half of the "Appendix B" should be moved to "5 Initial analysis and reference products".

2.3 Selected spatial and temporal domain P6L3: "in Sect. 0" might be "in Sect. 3.2" P6L4: "Additionally, seven clusters were set for an effective GNSS processing." It is unclear which areas in figure 1 are the "seven clusters", and what does "effective processing" mean. Please describe clearly. P6L6: "covered areas hit by" -> "covered areas were hit by"

2.4 Envisioned studies and activities P6L13: "real-time (RT)" -> "RT". Firstly appeared in P3L3. P6L14: "evaluating new analytical centres" -> Please explain what the "new analytical centres" are. P6L17: "real-time" -> "RT" P6L24:"IWV" -> "Integrated water vapor (IWV)"

3.1 Weather analysis, May 2013 P7L7: "daily accumulated precipitation (24-hour) from" -> "daily accumulated precipitation (24-hour) at Prague-Ruzyne (11518) synoptic station from" P7L11: "weak gradient at 500hPa" -> Of what "gradient"? P7L11-L24: Overall synoptic characteristics in May are described but no such description is seen for June. P7L21: "On May 31-31," -> Please check the exact date. P7L25: Figure 4 -> It would be nice if the area of the figure is more focusing on the area of the Benchmark campaign.

3.2 Extreme precipitation events in the Czech Republic, June 2013 P8L7: Figure 5 -> It would be nice if the distance scale and topography of the area are shown. Also, locations of "Šumava mountains", "Bohemia", "Plague", "Vltava", "Elve" should be pointed in the figure.

4.1 GNSS data P8L27: Figure 6. -> Mark for WVR station is hard to distinguish from that of radiosonde station. P9L3: "processing double-diference" -> It would be nice to add a reference on "double-difference", for example "Hoffmann-Wellenhof, et al, 2000: GPS theory and practice, 5th rev. ed., Springer-Verlag Wien New York." P9L5: Table 1 -> Does "Height" mean "Ellipsoidal height" or "Height above sea level (i.e. altitude)"?

P9L6: "All GNSS" -> It would be nice to add brief summary regarding antenna type (Choke ring, ground plane, etc.), and elevation cut off angle.

4.2 E-GVAP operational GNSS products P9L21:"14 analysis centres (and 29 solutuons)" -> It would be nice to show the names of 14 centres and 29 solutions in a table. P9L23:"TOUGH (2004)" -> Is it appropriate to use a project name rather than specific author(s) name?

4.4 NWM data and products P10L10: "Table 3" -> There is no information about geopotential height at each vertical layer. How users get geopotential height at each 3D grid point? P10L13: "Vertical resolution" -> "Vertical layers"

4.6 Water Vapour Radiometer data P10L28: "Water Vapor Radiometer (WVR)" -> WVR P11L4: "Integrated water vapor (IWV)" -> IWV P11L9: "GOPE and WTZR" -> Please describe the detail information about these two station.

5.2 NWM-derived tropospheric parameters P13L11:"German Research Centre for Geosciences (GFZ)" -> "GFZ" P13L26: "compared in the GOP-tropDB" -> It would be nice to explain how to correct height differences between NWM surface and GNSS antenna. PWV and ZTD are highly depend on height of antenna. It brings significant effect for the comparison. P14L2: "a negative mean bias of about 5mm" -> It is important to describe possible reasons for the large negative biases. P14L4: Figure 8 -> Color scale should be identical for both GNSS and NWM. P14L6: "As already seen in k in the local area model" -> What is "k"? P14L8: "good homogeneity" -> The expression is ambiguous. Please describe what it means by "homogeneity." P14L12: "23 times better horizontal resolution" -> Please explain of which horizontal resolution is "23 times better" than of which resolution. P14L15: "it has not been explained yet." -> At least, I would like authors to compare reproduced atmospheric fields among GFS, ERA-interim, and ALADIN. Which element field (surface pressure, water vapor, or temperature) is different in GFS from other two models?

5.3 GNSS and NWM tropospheric wet delay maps P14L22: "in Section 0" -> There is

no "section 0" in this article.

5.4 Comparison of horizontal gradients from GNSS and NWM P14L29: "zero a priori gradients" -> Please briefly explain this. I can't understand why "zero a priori gradients" leads "all solutions are considered as independent"? P15L9: "Figure 10" -> The amount of estimated gradients in GNSS analysis seems larger than those in NWM. Is this happened by chance? Did authors statistically compare gradients between GNSS and NWMs?

6 Conclusion P16L23-P17L2:"An initial study – GNSS4SWEC project." -> This paragraph seems to be a sudden. If authors want to discuss about the effects of hydrometeors, I want authors to discuss it in section 5 in association with difference in IWV (or ZWD) between GNSS analyses and NWMs.

Appendix A: GNSS tropospheric model A. 1 Mapping function coefficients – a, b, c A. 2 Horizontal tropospheric gradients Appendix B: Functional relation between NWM meteorological parameters and GNSS tropospheric model P20L1-P20L16:"For the Benchmark campaign – (%0.4)." -> This paragraph seems to be better discussing in section 5.

P22L13: "TOUGH" -> Is it appropriate to use a project name rather than specific author(s) name?

P24: Table 2. -> Available parameters of German stations are listed as "P, T, Td, RH". However, "Td" can be calculate using "P, T, and RH". Is there any reason why "Td" is listed only for Germany sites? P30: Figure 2. "June 4, 2013" -> "June 30, 2013" P32: Figure 4 -> It should be nice if the area of the figure is more focusing on the area of the Benchmark campaign. P33: Figure 5 -> It should be nice if the distance scale and topography of the area are shown. P34: Figure 6. -> Mark for WVR station is hard to distinguish from that of radiosonde station. P36: Figure 8 -> Color scale should be identical for both GNSS and NWM. Figure caption: "two numerical weather models" -> "three numerical weather models"

Please also note the supplement to this comment:
http://www.atmos-meas-tech-discuss.net/amt-2015-395/amt-2015-395-RC2-supplement.pdf
* * *

---

## Referee Comment (RC3) · Anonymous Referee #3 · 23 Mar 2016

This work is very interesting for both geodetic and meteorological communities in the framework of the ES1206 COST Action. The description of the data collected is very detailed and the data itself very completed having included radars, WVR data, radiosondes and synoptic data apart of several gnss products.

However, the test of the quality of GOP and GFZ gnss products by the comparison of the ZTD gnss with the three models described and also the subjective comparison of ZWD or horizontal gradients maps against the models could be more complete by using the other observations you already have for any of the case studies you describe. In

particular, in case of the comparison between the gnss ZTD vs the models, the results are difficult to understand if you don't study first how each model you present here performs in the case study, (if they are too humid or dry etc), because when comparing a product it should be with something you could trust ( or if not knowing why).

Maybe not so many explanations and maps describing the weather on these two months are necessary, just a general description of the period and then a detailed explanation of the case study chosen to perform the comparisons.

Very good job anyway, and complicated to accomplish, many data and long period. Differences of hydrometeors sounds very well for nowcasting purposes.

More comments:

2.2. P5,L:18: Numerical Weather Prediction models

2.3. P6, L: 1-5: Not the best words to describe the weather, (Mostly 'quiet', heavy 'raining', larger region instead of large...). And what is Sect 0 ?

3. P7, L2 and L5: Section 0 ?? About this section, probably not so many individual case studies explanations are needed, just the ones you are comparing afterwards, (like 31 may for example)..

3.1. Figure 2: The description on the text do not tell which stations the precipitation belongs to ( It is said on the legend of the figure but not on the text). Also X-axis could be clearer. Apart of this, a detailed explanation of the precipitations amounts is given for may and june, and this plot is not very clear to see it.

P7, L21: Typo May 31-31

P7,L25 and 26: Why aren't they inside the next section 3.2 if they are referred to june ?

3.2 P8, L 5-10_ 'Professional 'meteorological stations... are they referred to synop stations? Do the stations give the feeling on a rain episode?..

4.1. Table 1, brief explanation may be needed in the text.

4.3. Table 2, brief explanation may be needed in the text.

4.4. When you describe ALADIN-CZ model you don't explain if it has Data assimilation, what kind of D.A. and which data it assimilates. Better description of the data assimilation (especially sources of humidity) of the models used may be needed, so it may be easier to explain afterwards the biases/SD with one model or other.

5.2 P14, L12, you say Table 1, isn't it 5? It could be interesting doing this comparison also with radiosonde (RS) data and gnss sites collocated with radiosondes: model-RS-gnss.

5.3. P14, L 22; Section 0 again.

Figure 9: It could be interesting compare these figure of ZWD of 31 may with another image, radar, satellite WV, or even accumulated precipitation the day after, that permit to do any other validation of this maps, looking to any correspondence with real data.

6. P16, L23: This part could be inside section 5, and not in the conclusions.

---

## Author Comment (AC1) · 29 Apr 2016

**REVIEW NUMBER 1**

*This is a very fine overview article, providing a short overview of the GNSS4SWEC ES 1206 Cost action, an extensive, detailed overview of the selection and handling of data for the GNSS4SWEC benchmark campaign, and some first results from the inter comparisons of GNSS and NWP derived ZTDs and ZTD gradients.*

*It is clear the benchmark data have been carefully screened and processed, providing a very useful dataset for further studies. Among the most interesting results are that GOP and GFZ results agree well (network versus ppp), that higher resolution NWP seems ot agree better with the GNSS, and that GFS NWP is biased with respect to the other solutions.*

*The the NWP community the finding of significantly larger gradients from the GNSS processing than found in NWP is very interesting, and potentially useful as an added data source. Likewise the finding that in certain situations the contribution from liquid water and ice is not negligible (estimated from NWP based calculations).*

*In my view the manuscript is almost ready for acceptance. I would recommend correct points 1 and 3 below before acceptance, while point 2 might be better suited by an online comment from the authors, given that this type of manuscript does not contain many details.*

*A few more detailed comments:*

1) Many places in the text a reference to section 0 is given. Presumably that's the annex (judging from the numbering of the equations there), but the annex has no number.

*Manuscript changed (all cases corrected: Sect 2.3, Sect 3 and Sect 5.3).*

2) A bias for GFS ZTD is found. There could be several reasons, some of which are related to how the numerical integrator for NWP ZTD is made, regarding both interpolation (different vertical resolutions) and extrapolation (top of the model).

*Manuscript changed (Sect 5.2 completed). A possible explanation for the systematic deviation between NCEP's GFS and ECMWF's ERA-Interim ZTDs is the low vertical resolution of the NCEP GFS data (available on 26 pressure levels). In fact, the bias in the ZTD stems from a bias in the ZWD. For a comparison between all the NCEP GFS and ECMWF ERA-Interim tropospheric parameters see Zus et al. 2015 ('WG1 model sub-group summary', ES1206-GNSS4WEC COST Meeting, Wrozlaw, September 28 – October 1, 2015). Note that a comparable bias between NCEP and ECMWF ZWDs was reported by Urquhart et al. 2011 ('Generation and Assessment of VMF1-Type Grids using North-American Numerical Weather Models', presented at XXV IUGG General Assembly, Melbourne, Australia, June 28th – July 7th, 2011, available at [http://unbvmf1.gge.unb.ca/Publications.html](http://unbvmf1.gge.unb.ca/Publications.html)). We also note that the interpolation routine, that is used to compute the refractivity at arbitrary points, is the same for both NWMs. Therefore the low vertical resolution of the NCEP GFS data also implies larger interpolation errors.*

3) Equation 0.6, for the effect of the hydro meteors, contains numbers on the right hand side, which seem to call for units. Is what is called "mass content" of hydro meteors another word for their mass density?

*Manuscript changed (more clear explanation added, Sect 5.5). $M_{lw}$ is the mass content per unit of air volume of liquid water hydrometeors (e.g. cloud water and rainwater) and $M_{ice}$ the mass content per unit of air volume of icy hydrometeors (e.g. pristine ice, snow and graupel).*

Speaking about units there is on the other hand no need to provide units for the density, pressure, temperature and gas constants further up page 19, as long as no values are given for the constants in equations 0.4 and 0.5.

*Manuscript changed (units deleted from equations in Sect 5)*

The idea to include the hydro meteor contribution in the NWP cost function (if the NWP contains the relevant parameters) mentioned in the conclusion is interesting. In reality it is very complicated, since they are normally not variables in the same sense as specific humidity of the NWP, and the NWP can be far of regarding their size. But pointing out that hydro meteors do sometimes contribute significantly to what appears otherwise to be ZWD is important.

*Thank you for this comment.*

---

## Author Comment (AC2) · 29 Apr 2016

**REVIEW NUMBER 2**

Specifications and characteristics of the Benchmark campaign data set are described in detail. I believe the data set will contribute for both geodesy and meteorology in not only Europe but other countries and regions. However, there are not a few typos, insufficient explanations in body, tables, and figures. I would like authors to carefully polish the article. Followings are for reference.

About the chapter arrangement:

(1) It would be nice to move "3. Case study episode in 2013" to after "4. Benchmark data set" and before "5. Initial analysis and reference products"

Manuscript changed (order of the chapters has been changed).

(2) Effects of hydrometeor described in the latter half of the "6. Conclusion" and latter half of the "Appendix B" should be moved to "5. Initial analysis and reference products".

Manuscript changed (Appendix B has been removed). The effect of hydrometeors is now presented in Sect 5.5. Conclusion has been modified referring results about hydrometeors shown in Sect 5.5.

**Abstract.**

**1 Introduction**

**2 GNSS4SWEC Benchmark campaign**

2.1 Description of WG1 objectives

**2.2 Data inventory and requirements for the Benchmark design**

2.3 Selected spatial and temporal domain

P6L3: "in Sect. 0" might be "in Sect. 3.2"

**Manuscript changed (Sect 3.2).**

P6L4: "Additionally, seven clusters were set for an effective GNSS processing." It is unclear which areas in figure 1 are the "seven clusters", and what does "effective processing" mean. Please describe clearly.

Manuscript changed (the text was reformatted to better describe the situation). Additionally, the whole domain was geographically divided into nine clusters (five within the 'core' domain and four within the 'extended' domain, see different colors of individual station in Figure 1) to allow reasonable GNSS data handling.

P6L6: "covered areas hit by" -> "covered areas were hit by"

**Manuscript changed.**

**2.4 Envisioned studies and activities**

P6L13: "real-time (RT)" -> "RT". Firstly appeared in P3L3.

**Manuscript changed.**

P6L14: "evaluating new analytical centres" > Please explain what the "new analytical centres" are.

Manuscript changed (text was slightly reformatted). New analytical centers were established with support of the GNSS4SWEC COST project in Turkey, Bulgaria, Greece and Iceland. The Benchmark dataset could be used for evaluating own built solutions while comparing them with solutions from well-established centers.

P6L17: "real-time" -> "RT"

Manuscript changed.

P6L24:"IWV" -> "Integrated water vapor (IWV)"

Manuscript changed.

**3 Case study episodes in 2013**

3.1 Weather analysis, May 2013

P7L7: "daily accumulated precipitation (24-hour) from" -> "daily accumulated precipitation (24-hour) at Prague-Ruzyne (11518) synoptic station from"

Manuscript changed (correction accepted).

P7L11: "weak gradient at 500hPa" -> Of what "gradient"?

Manuscript changed (weak gradients of the geopotential). On May 5, the precipitation was associated with an upper level trough and weak gradients of the geopotential at 500 hPa.

P7L11-L24: Overall synoptic characteristics in May are described but no such description is seen for June.

We believe a sufficient description is given in Sect 4.2.

P7L21: "On May 31-31," -> Please check the exact date.

Manuscript changed (corrected: May 30 – 31).

P7L25: Figure 4 -> It would be nice if the area of the figure is more focusing on the area of the Benchmark campaign.

Manuscript changed (Figure 4 edited). For a good overview on the synoptic pattern it is better to look at the whole Europe region and see this big process over the Benchmark region. Nevertheless, both pictures where zoomed to show better central Europe.

**3.2 Extreme precipitation events in the Czech Republic, June 2013**

P8L7: Figure 5 -> It would be nice if the distance scale and topography of the area are shown. Also, locations of "Šumava mountains", "Bohemia", "Plague", "Vltava", "Elve" should be pointed in the figure.

Manuscript changed (Figure 5 edited). Geographic names were not included since it would deteriorate the legibility of the figure. Legends for x/y axes with geographic coordinates and a distance scales were added.

**4 Benchmark data set**

**4.1 GNSS data**

P8L27: Figure 6. -> Mark for WVR station is hard to distinguish from that of radiosonde station.

Manuscript changed (symbols for WVR and RS in Figure 6 were changed).

P9L3: "processing double-diference" -> It would be nice to add a reference on "double-difference", for example "Hoffmann-Wellenhof, et al, 2000: GPS theory and practice, 5th rev. ed., Springer-Verlag Wien New York."

Manuscript changed (reference added).

P9L5: Table 1 -> Does "Height" mean "Ellipsoidal height" or "Height above sea level (i.e. altitude)"?

Manuscript changed (ellipsoidal height is correct).

P9L6: "All GNSS" -> It would be nice to add brief summary regarding antenna type (Choke ring, ground plane, etc.), and elevation cut off angle.

It is almost impossible to provide such information since practically all types of receivers/antennas were used. Data from each country came from at least one stand-alone network and also within these networks different equipment is used at individual stations. So a real mixture of various devices exists there.

**4.2 E-GVAP operational GNSS products**

P9L21:"14 analysis centres (and 29 solutions)" -> It would be nice to show the names of 14 centres and 29 solutions in a table.

From our perspective this is out of scope of the paper while E-GVAP operational products are only supplementary to the whole dataset.

P9L23:"TOUGH (2004)" -> Is it appropriate to use a project name rather than specific author(s) name?

Manuscript changed (TOUGH Final report by H. Vedel referenced in Sect 3.2).

**4.3 Synoptic data**

**4.4 NWM data and products**

P10L10: "Table 3" -> There is no information about geopotential height at each vertical layer. How users get geopotential height at each 3D grid point?

Manuscript changed (added text in Sec 3.4). Geopotential heights are provided for the model surface only while the model levels are expressed using hybrid vertical coordinates. The atmospheric pressure between model levels is calculated with two pre-defined coefficients (a,b), reference pressure (101325Pa) and top level pressure, all provided in each file. Geopotential heights at any level could be additionally calculated using the hypsometric formula, atmospheric pressure and temperature at individual model levels.

P10L13: "Vertical resolution" -> "Vertical layers"

**Manuscript changed.**

**4.5 Radiosonde data**

**4.6 Water Vapour Radiometer data**

P10L28: "Water Vapor Radiometer (WVR)" -> WVR

Manuscript changed.

P11L4: "Integrated water vapor (IWV)" -> IWV

**Manuscript changed.**

P11L9: "GOPE and WTZR" -> Please describe the detail information about these two stations.

We don't think it would be useful to describe instruments at GOPE and WTZR since no data from them were available for the Benchmark dataset. We wanted to mention that although in the area there were two other WVRs in hope to be used, but were unfortunately, out of order during the Benchmark time period.

**4.7 Meteorological Radar images**

**5 Initial analysis and reference products**

5.1 Reference tropospheric products

**5.2 NWM-derived tropospheric parameters**

P13L11:"German Research Centre for Geosciences (GFZ)" -> "GFZ"

**Manuscript changed.**

P13L26: "compared in the GOP-TropDB" -> It would be nice to explain how to correct height differences between NWM surface and GNSS antenna. PWV and ZTD are highly dependent on height of antenna. It brings significant effect for the comparison.

Manuscript changed (added text in Sect 5.2). The same vertical parameter scaling is used in the GOP-TropDB if there is a need to compare parameters at two collocated stations with different heights, e.g. GNSS vs. other space geodetic techniques. This procedure is however not applied in case of our GNSS vs. NWM comparison since NWM parameters are calculated for the GNSS locations and heights. P14L2: "a negative mean bias of about 5mm" -> It is important to describe possible reasons for the large negative biases.

Manuscript changed (an explanation added in the text). "A possible explanation for the systematic deviation between NCEP's GFS and ECMWF's ERA-Interim ZTDs is the low vertical resolution of the NCEP GFS data (available on 26 pressure levels). In fact, the bias in the ZTD stems from a bias in ZWD. For a comparison between all NCEP GFS and ECMWF ERA-Interim tropospheric parameters, see Zus et al 2015 ('WG1 model sub-group summary', ES1206-GNSS4WEC COST Meeting, Wroclaw, Sept 28 – Oct 1, 2015).

Note that a comparable bias between NCEP and ECMWF ZWDs was reported by Urquhart et al. 2011 ('Generation and Assessment of VMF1-Type Grids using North-American Numerical Weather Models', presented at XXV IUGG General Assembly, Melbourne, Australia, June 28th – July 7th, 2011, available at http://unbvmf1.qqe.unb.ca/Publications.html). We also note that the interpolation routine, that is used to compute the refractivity at arbitrary points, is the same for both NWMs. Therefore the low vertical resolution of the NCEP GFS data also implies larger interpolation errors."

P14L4: Figure 8 -> Color scale should be identical for both GNSS and NWM.

Color scales are identical for all three comparisons of GNSS vs. NWM, just different scales are used for biases (left plots) and standard deviations (right plots) since the standard deviation can't be negative.

P14L6: "As already seen in k in the local area model" -> What is "k"?

Manuscript changed (there was a typo, the reference should have pointed to Table 5).

P14L8: "good homogeneity" -> The expression is ambiguous. Please describe what it means by "homogeneity."

Manuscript changed (we use the word agreement instead).

P14L12: "23 times better horizontal resolution" -> Please explain of which horizontal resolution is "23 times better" than of which resolution.

Manuscript changed (sentence reformatted). "This shows that a complex terrain such as in the Alps is much better captured by the meso-scale model ALADIN-CZ with up to 23 times better horizontal resolution than both used global models have."

P14L15: "it has not been explained yet." -> At least, I would like authors to compare reproduced atmospheric fields among GFS, ERA-interim, and ALADIN. Which element field (surface pressure, water vapor, or temperature) is different in GFS from other two models?

Manuscript changed (see P14L2 comment). The difference stems from ZWD, i.e. the water vapor and/or temperature field. We will certainly not start to compare atmospheric fields in this manuscript which is out of its scope.

**5.3 GNSS and NWM tropospheric wet delay maps**

P14L22: "in Section 0" -> There is no "section 0" in this article.

Manuscript changed (Sect 3.2 corrected).

**5.4 Comparison of horizontal gradients from GNSS and NWM**

P14L29: "zero a priori gradients" -> Please briefly explain this. I can't understand why "zero a priori gradients" leads "all solutions are considered as independent"?

Manuscript changed (text reformatted). "No information about gradients was introduced in the GOP or GFZ reference GNSS analyses and thus they can be considered as fully independent from NWM derived gradients."

P15L9: "Figure 10" -> The amount of estimated gradients in GNSS analysis seems larger than those in NWM. Is this happened by chance? Did authors statistically compare gradients between GNSS and NWMs?

No. Current thinking is that the low horizontal resolution of the NWMs (1 by 1 degree) is responsible for the underestimation of gradients. Therefore we added the following comment in the manuscript: "For example, Zus et al. 2016 ('Station specific NWM based tropo parameters for the Benchmark campaign', ES1206-GNSS4WEC COST Workshop, Iceland, March 8-10, 2016) show how an increased horizontal resolution of the NWM amplifies the gradient components under severe weather conditions."

**6 Conclusion**

P16L23-P17L2:"An initial study – GNSS4SWEC project." -> This paragraph seems to be a sudden. If authors want to discuss about the effects of hydrometeors, I want authors to discuss it in section 5 in association with difference in IWV (or ZWD) between GNSS analyses and NWMs.

Manuscript changed (the effect of hydrometeors is now presented in Sect 5.5). A discussion about the overestimation of ZWD and IWV from GNSS induced by such hydrometeors contribution is presented in this new section. Conclusion has been modified referring results about hydrometeors shown in Sect 5.5.

Appendix A: GNSS tropospheric model

A. 1 Mapping function coefficients – a, b, c

A. 2 Horizontal tropospheric gradients

Appendix B: Functional relation between NWM meteorological parameters and GNSS tropospheric

model

P20L1-P20L16:"For the Benchmark campaign – ((0.4)." -> This paragraph seems to be better discussing in section 5.

Manuscript changed (the effect of hydrometeors is now presented in Sect 5.5).

P22L13: "TOUGH" -> Is it appropriate to use a project name rather than specific author(s) name?

Manuscript changed (changed to TOUGH Final report by H. Vedel referenced in Sect 3.2).

P24: Table 2. -> Available parameters of German stations are listed as "P, T, Td, RH". However,

"Td" can be calculated using "P, T, and RH". Is there any reason why "Td" is listed only for Germany sites?

Manuscript changed ("Td" was removed from the line for Germany).

P30: Figure 2. "June 4, 2013" -> "June 30, 2013"

Manuscript changed (Figure 2 edited). We added the name of the station on the text and we changed the x axis, putting on it more days for clarifying.

P32: Figure 4 -> It should be nice if the area of the figure is more focusing on the area of the Benchmark campaign.

Manuscript changed (Figure 4 edited).

P33: Figure 5 -> It should be nice if the distance scale and topography of the area are shown.

Manuscript changed (Figure 5 edited).

P34: Figure 6. -> Mark for WVR station is hard to distinguish from that of radiosonde station.

Manuscript changed (Figure 6 edited).

P36: Figure 8 -> Color scale should be identical for both GNSS and NWM.

Color scales are identical for all three comparisons of GNSS vs. NWM, just different scales are used for biases (left plots) and standard deviations (right plots) since the standard deviation can't become negative.

Figure caption: "two numerical weather models" -> "three numerical weather models"

Manuscript changed (Figure caption corrected, three models).